# TOWARDS A BETTER THEORETICAL UNDERSTANDING OF INDEPENDENT SUBNETWORK TRAINING

## ABSTRACT

Modern advancements in large-scale machine learning would be impossible without the paradigm of data-parallel distributed computing. Since distributed computing with large-scale models imparts excessive pressure on communication channels, significant recent research has been directed toward co-designing communication compression strategies and training algorithms with the goal of reducing communication costs. While pure data parallelism allows better data scaling, it suffers from poor model scaling properties. Indeed, compute nodes are severely limited by memory constraints, preventing further increases in model size. For this reason, the latest achievements in training giant neural network models also rely on some form of model parallelism. In this work, we take a closer theoretical look at Independent Subnetwork Training (IST), which is a recently proposed and highly effective technique for solving the aforementioned problems. We identify fundamental differences between IST and alternative approaches, such as distributed methods with compressed communication, and provide a precise analysis of its optimization performance on a quadratic model.

## 1 INTRODUCTION

A huge part of today's machine learning success is driven by the possibility of building more and more complex models and training them on increasingly larger datasets. This rapid progress has become feasible due to advancements in distributed optimization, which is necessary for proper scaling when the size of the training data grows (Zinkevich et al., 2010). In a typical scenario, data parallelism is used for efficiency and implies sharding the dataset across computing devices. This allowed very efficient scaling and acceleration of training moderately sized models by using additional hardware (Goyal et al., 2018). However, this data parallel approach can suffer from communication bottleneck, which has sparked extensive research on distributed optimization with compressed communication of the parameters between nodes (Alistarh et al., 2017; Konečný et al., 2016; Seide et al., 2014).

### 1.1 THE NEED FOR MODEL PARALLELISM

Despite its efficiency, data parallelism has some fundamental limitations when it comes to scaling up the model size. As the dimensions of a model increase, the amount of memory required to store and update the parameters also increases, which becomes problematic due to resource constraints on individual devices. This has led to the development of model parallelism (Dean et al., 2012; Richtárik & Takáč, 2016), which splits a large model across multiple nodes, with each node responsible for computations of parts of the model (Farber & Asanovic, 1997; Zhang et al., 1989). However, naive model parallelism also poses challenges because each node can only update its portion of the model based on the data it has access to. This creates a need for very careful management of communication between devices. Thus, a combination of both data and model parallelism is often necessary to achieve efficient and scalable training of huge models.

**IST.** Independent Subnetwork Training (IST) is a technique that suggests dividing a neural network into smaller subparts, training them in a distributed parallel fashion, and then aggregating the results to update the weights of the whole model. In IST, every subnetwork can operate independently and has fewer parameters than the full model, which not only reduces the load on computing nodes but also results in faster synchronization. A generalized analog of the described method is formalized

---

**Algorithm 1** Distributed Submodel (Stochastic) Gradient Descent

---

1: **Parameters:** learning rate $\gamma > 0$; sketches $\mathbf{C}_1, \ldots, \mathbf{C}_n$; initial model $x^0 \in \mathbb{R}^d$
2: **for** $k = 0, 1, 2 \ldots$ **do**
3:     Select submodels $w_i^k = \mathbf{C}_i^k x^k$ for $i \in [n]$ and broadcast to all computing nodes
4:     **for** $i = 1, \ldots, n$ in parallel **do**
5:         Compute local (stochastic) gradient w.r.t. submodel: $\mathbf{C}_i^k \nabla f_i(w_i^k)$
6:         Take (maybe multiple) gradient descent step $z_i^+ = w_i^k - \gamma \mathbf{C}_i^k \nabla f_i(w_i^k)$
7:         Send $z_i^+$ to the server
8:     **end for**
9:     Aggregate/merge received submodels: $x^{k+1} = \frac{1}{n} \sum_{i=1}^n z_i^+$
10: **end for**

---

as an iterative procedure in Algorithm 1. This paradigm was pioneered by Yuan et al. (2022) for networks with fully connected layers and was later extended to ResNets (Dun et al., 2022) and Graph architectures (Wolfe et al., 2021). Previous experimental studies have shown that IST is a very promising approach for various applications as it allows to effectively combine data and model parallelism and train larger models with limited compute. In addition, Liao & Kyrillidis (2022) performed theoretical analysis of IST for overparameterized single hidden layer neural networks with ReLU activations. The idea of IST was also recently extended to the federated setting via an asynchronous distributed dropout technique (Dun et al., 2023).

**Federated Learning.** Another important setting when the data is distributed (due to privacy reasons) is Federated Learning (Kairouz et al., 2021; Konečný et al., 2016; McMahan et al., 2017). In this scenario, computing devices are often heterogeneous and more resource-constrained (Caldas et al., 2018) (e.g. mobile phones) in comparison to data-center settings. Such challenges have prompted extensive research efforts into selecting smaller and more efficient submodels for local on-device training (Alam et al., 2022; Charles et al., 2022; Chen et al., 2022; Diao et al., 2021; Horvath et al., 2021; Jiang et al., 2022; Lin et al., 2022; Qiu et al., 2022; Wen et al., 2022; Yang et al., 2022). Many of these works propose approaches to adapt submodels, often tailored to specific neural network architectures, based on the capabilities of individual clients for various machine learning tasks. However, there is a lack of comprehension regarding the theoretical properties of these methods.

### 1.2 SUMMARY OF CONTRIBUTIONS

After reviewing the literature, we found that a rigorous understanding of IST convergence is virtually non-existent, which motivated this work. The main contributions of this paper include

• A novel approach to analyzing distributed methods that combine data and model parallelism by operating with sparse submodels for a quadratic model.

• The first analysis of independent subnetwork training in homogeneous and heterogeneous scenarios without restrictive assumptions on gradient estimators.

• Identification of the settings when IST can optimize very efficiently or not converge to the optimal solution but only to an irreducible neighborhood that is also tightly characterized.

• Experimental validation of the proposed theory through carefully designed illustrative experiments. The results, together with all the proofs, are given in the Appendix.

## 2 FORMALISM AND SETUP

We consider the standard optimization formulation of distributed learning (Wang et al., 2021)

$$\min_{x \in \mathbb{R}^d} \left[ f(x) := \frac{1}{n} \sum_{i=1}^n f_i(x) \right], \tag{1}$$

where $n$ is the number of clients/workers, and each $f_i : \mathbb{R}^d \to \mathbb{R}^d$ represents the loss of the model parameterized by vector $x \in \mathbb{R}^d$ on the data of client $i$.

A typical Stochastic Gradient Descent (SGD)-type method for solving this problem has the form

$$x^{k+1} = x^k - \gamma g^k, \qquad g^k = \tfrac{1}{n} \sum_{i=1}^{n} g_i^k, \tag{2}$$

where $\gamma > 0$ is the stepsize and $g_i^k$ is a suitably constructed estimator of $\nabla f_i(x^k)$. In the distributed setting, computation of gradient estimators $g_i^k$ is typically performed by clients, and the results are sent to the server, which subsequently performs aggregation via averaging $g^k = \tfrac{1}{n} \sum_{i=1}^{n} g_i^k$. The average is then used to update the model $x^{k+1}$ via a gradient-type method (2), and at the next iteration, the model is broadcasted back to the clients. The process is repeated iteratively until a suitable model is found.

One of the main techniques used to accelerate distributed training is lossy *communication compression* (Alistarh et al., 2017; Konečný et al., 2016; Seide et al., 2014), which suggests applying a (possibly randomized) lossy compression mapping $\mathcal{C}$ to a vector/matrix/tensor $x$ before broadcasting. This reduces the bits sent per communication round at the cost of transmitting a less accurate estimate $\mathcal{C}(x)$ of $x$. Described technique can be formalized in the following definition.

**Definition 1** (Unbiased compressor). *A randomized mapping $\mathcal{C} : \mathbb{R}^d \to \mathbb{R}^d$ is an **unbiased compression operator** ($\mathcal{C} \in \mathbb{U}(\omega)$ for brevity) if for some $\omega \geq 0$ and $\forall x \in \mathbb{R}^d$*

$$\mathbb{E}\left[\mathcal{C}(x)\right] = x, \qquad \mathbb{E}\left[\|\mathcal{C}(x) - x\|^2\right] \leq \omega \|x\|^2. \tag{3}$$

A notable example of a mapping from this class is the *random sparsification* (Rand-q for $q \in [d] := \{1, \ldots, d\}$) operator defined by

$$\mathcal{C}_{\text{Rand-q}}(x) := \mathbf{C}_q x = \frac{d}{q} \sum_{i \in S} e_i e_i^\top x, \tag{4}$$

where $e_1, \ldots, e_d \in \mathbb{R}^d$ are standard unit basis vectors in $\mathbb{R}^d$, and $S$ is a random subset of $[d]$ sampled from the uniform distribution on the all subsets of $[d]$ with cardinality $q$. Rand-q belongs to $\mathbb{U}\left(d/q - 1\right)$, which means that the more elements are "dropped" (lower $q$), the higher the variance $\omega$ of the compressor.

In this work, we are mainly interested in a more general class of operators than mere sparsifiers. In particular, we are interested in compressing via the application of random matrices, i.e., via *sketching*. A sketch $\mathbf{C}_i^k \in \mathbb{R}^{d \times d}$ can be used to represent submodel computations in the following way:

$$g_i^k := \mathbf{C}_i^k \nabla f_i(\mathbf{C}_i^k x^k), \tag{5}$$

where we require $\mathbf{C}_i^k$ to be a symmetric positive semi-definite matrix. Such gradient estimates correspond to computing the local gradient with respect to a sparse submodel $\mathbf{C}_i^k x^k$, and additionally sketching the resulting gradient with the same matrix $\mathbf{C}_i^k$ to guarantee that the resulting update lies in the lower-dimensional subspace.

Using this notion, Algorithm 1 (with 1 local gradient step) can be represented in the following form:

$$x^{k+1} = \tfrac{1}{n} \sum_{i=1}^{n} \left[\mathbf{C}_i^k x^k - \gamma \mathbf{C}_i^k \nabla f_i(\mathbf{C}_i^k x^k)\right], \tag{6}$$

which is equivalent to the SGD-type update (2) when the *perfect reconstruction* property holds (with probability one)

$$\mathbf{C}^k := \tfrac{1}{n} \sum_{i=1}^{n} \mathbf{C}_i^k = \mathbf{I},$$

where $\mathbf{I}$ is the identity matrix. This property is inherent for a specific class of compressors that are particularly useful for capturing the concept of an independent subnetwork *partition*.

**Definition 2** (Permutation sketch). *Assume that $d \geq n$ and $d = qn$, where $q \geq 1$ is an integer[1]. Let $\pi = (\pi_1, \ldots, \pi_d)$ be a random permutation of $[d]$. Then for all $x \in \mathbb{R}^d$ and each $i \in [n]$, we define* Perm-q *operator*

$$\mathbf{C}_i := n \cdot \sum_{j=q(i-1)+1}^{qi} e_{\pi_j} e_{\pi_j}^\top. \tag{7}$$

---

[1] While this condition may look restrictive, it naturally holds for distributed learning in a data-center setting. Permutation sparsifiers were introduced by Szlendak et al. (2022) and generalized to other scenarios (like $n \geq d$).

`Perm-q` is unbiased and can be conveniently used for representing a structured decomposition of the model, such that every client $i$ is responsible for computations over a submodel $\mathbf{C}_i x^k$.

Our convergence analysis relies on the assumption that was previously used for coordinate descent-type methods.

**Assumption 1** (Matrix smoothness). *A differentiable function $f : \mathbb{R}^d \to \mathbb{R}$ is $\mathbf{L}$-smooth, if there exists a positive semi-definite matrix $\mathbf{L} \in \mathbb{R}^{d \times d}$ such that*

$$f(x + h) \leq f(x) + \langle \nabla f(x), h \rangle + \tfrac{1}{2} \langle \mathbf{L}h, h \rangle, \qquad \forall x, h \in \mathbb{R}^d. \tag{8}$$

A standard $L$-smoothness condition is obtained as a special case of (8) for $\mathbf{L} = L \cdot \mathbf{I}$. Matrix smoothness was previously used for designing data-dependent gradient sparsification to accelerate optimization in communication-constrained settings (Safaryan et al., 2021; Wang et al., 2022).

### 2.1  ISSUES WITH EXISTING APPROACHES

Consider the simplest gradient descent method with a compressed model in the single-node setting:

$$x^{k+1} = x^k - \gamma \nabla f(\mathcal{C}(x^k)). \tag{9}$$

Algorithms belonging to this family require a different analysis in comparison to SGD (Gorbunov et al., 2020; Gower et al., 2019), Distributed Compressed Gradient Descent (Alistarh et al., 2017; Khirirat et al., 2018), and Randomized Coordinate Descent (Nesterov, 2012; Richtárik & Takáč, 2014)-type methods because the gradient estimator is no longer unbiased

$$\mathbb{E}\left[\nabla f(\mathcal{C}(x))\right] \neq \nabla f(x) = \mathbb{E}\left[\mathcal{C}(\nabla f(x))\right]. \tag{10}$$

This is why such kind of algorithms (9) are harder to analyze. So, prior results for *unbiased* SGD (Khaled & Richtárik, 2023) cannot be directly reused. Furthermore, the nature of the bias in this type of gradient estimator does not exhibit additive noise, thereby preventing the application of previous analyses for biased SGD (Ajalloeian & Stich, 2020).

An assumption like the bounded stochastic gradient norm extensively used in previous works (Lin et al., 2019; Zhou et al., 2022) hinders an accurate understanding of such methods. This assumption hides the fundamental difficulty of analyzing a biased gradient estimator:

$$\mathbb{E}\left[\|\nabla f(\mathcal{C}(x))\|^2\right] \leq G \tag{11}$$

and may not hold, even for quadratic functions $f(x) = x^\top \mathbf{A} x$. In addition, in the distributed setting, such a condition can result in vacuous bounds (Khaled et al., 2020) as it does not capture heterogeneity accurately.

### 2.2  SIMPLIFICATIONS TAKEN

To conduct a thorough theoretical analysis of methods that combine data with model parallelism, we simplify the algorithm and problem setting to isolate the unique effects of this approach. The following considerations are made:

- **(1)** We assume that every node $i$ computes the true gradient at the submodel $\mathbf{C}_i \nabla f_i(\mathbf{C}_i x^k)$.
- **(2)** A notable difference compared to the original IST Algorithm 1 is that workers perform a single gradient descent step (or just gradient computation).
- **(3)** Finally, we consider a special case of a quadratic model (12) as a loss function (1).

Condition **(1)** is mainly for the sake of simplicity and clarity of exposition and can be potentially generalized to stochastic gradient computations. Condition **(2)** is imposed because local steps did not bring any theoretical efficiency improvements for heterogeneous settings until very recently (Mishchenko et al., 2022), and even then, only with the introduction of additional control variables, which goes against the requirements of resource-constrained device settings. The reason behind **(3)** is that despite its apparent simplicity, the quadratic problem has been used extensively to study properties of neural networks (Zhang et al., 2019; Zhu et al., 2022). Moreover, it is a non-trivial

model, which makes it possible to understand complex optimization algorithms (Arjevani et al., 2020; Cunha et al., 2022; Goujaud et al., 2022). The quadratic problem is suitable for observing complex phenomena and providing theoretical insights, which can also be observed in practical scenarios.

Having said that, we consider a special case of problem (1) for symmetric matrices $\mathbf{L}_i$

$$f(x) = \frac{1}{n} \sum_{i=1}^n f_i(x), \qquad f_i(x) \equiv \frac{1}{2} x^\top \mathbf{L}_i x - x^\top \mathrm{b}_i \,. \tag{12}$$

In this case, $f(x)$ is $\overline{\mathbf{L}}$-smooth, and $\nabla f(x) = \overline{\mathbf{L}} x - \overline{\mathrm{b}}$, where $\overline{\mathbf{L}} = \frac{1}{n} \sum_{i=1}^n \mathbf{L}_i$ and $\overline{\mathrm{b}} := \frac{1}{n} \sum_{i=1}^n \mathrm{b}_i$.

## 3 RESULTS IN THE INTERPOLATION CASE

First, let us examine the case of $\mathrm{b}_i \equiv 0$, which we call interpolation for quadratics, and perform the analysis for general sketches $\mathbf{C}_i^k$. In this case, the gradient estimator (2) takes the form

$$g^k = \frac{1}{n} \sum_{i=1}^n \mathbf{C}_i^k \nabla f_i(\mathbf{C}_i^k x^k) = \frac{1}{n} \sum_{i=1}^n \mathbf{C}_i^k \mathbf{L}_i \mathbf{C}_i^k x^k = \overline{\mathbf{B}}^k x^k \tag{13}$$

where $\overline{\mathbf{B}}^k := \frac{1}{n} \sum_{i=1}^n \mathbf{C}_i^k \mathbf{L}_i \mathbf{C}_i^k$. We prove the following result for a method with such an estimator.

**Theorem 1.** *Consider the method* (2) *with estimator* (13) *for a quadratic problem* (12) *with* $\overline{\mathbf{L}} \succ 0$ *and* $\mathrm{b}_i \equiv 0$. *Then if* $\overline{\mathbf{W}} := \frac{1}{2} \mathbb{E}\left[ \overline{\mathbf{L}} \overline{\mathbf{B}}^k + \overline{\mathbf{B}}^k \overline{\mathbf{L}} \right] \succeq 0$ *and there exists a constant* $\theta > 0$:

$$\mathbb{E}\left[ \overline{\mathbf{B}}^k \overline{\mathbf{L}} \overline{\mathbf{B}}^k \right] \preceq \theta \overline{\mathbf{W}}, \tag{14}$$

*and the step size is chosen as* $0 < \gamma \leq \frac{1}{\theta}$, *the iterates satisfy*

$$\frac{1}{K} \sum_{k=0}^{K-1} \mathbb{E}\left[ \left\| \nabla f(x^k) \right\|_{\overline{\mathbf{L}}^{-1} \overline{\mathbf{W}} \overline{\mathbf{L}}^{-1}}^2 \right] \leq \frac{2\left( f(x^0) - \mathbb{E}[f(x^K)] \right)}{\gamma K}, \tag{15}$$

$$\mathbb{E}\left[ \|x^k - x^\star\|_{\overline{\mathbf{L}}}^2 \right] \leq \left( 1 - \gamma \lambda_{\min}\left( \overline{\mathbf{L}}^{-\frac{1}{2}} \overline{\mathbf{W}} \overline{\mathbf{L}}^{-\frac{1}{2}} \right) \right)^k \|x^0 - x^\star\|_{\overline{\mathbf{L}}}^2. \tag{16}$$

This theorem establishes an $\mathcal{O}(1/K)$ convergence rate with a constant step size up to a stationary point and linear convergence for the expected distance to the optimum $x^\star := \arg\min f(x)$. Note that we employ weighted norms in our analysis, as the considered class of loss functions satisfies the matrix $\overline{\mathbf{L}}$-smoothness Assumption 1. The use of standard Euclidean distance may result in loose bounds that do not recover correct rates for special cases like gradient descent.

It is important to highlight that the inequality (14) may not hold (for any $\theta > 0$) in the general case as the matrix $\overline{\mathbf{W}}$ is not guaranteed to be positive (semi-)definite in the case of general sampling. The intuition behind this issue is that arbitrary sketches $\mathbf{C}_i^k$ can result in the gradient estimator $g^k$, which is misaligned with the true gradient $\nabla f(x^k)$. Specifically, the inner product $\langle \nabla f(x^k), g^k \rangle$ can be negative, and there is no expected descent after one step.

Next, we give examples of samplings for which the inequality (14) can be satisfied.

**1. Identity.** Consider $\mathbf{C}_i \equiv \mathbf{I}$. Then $\overline{\mathbf{B}}^k = \overline{\mathbf{L}}$, $\overline{\mathbf{B}}^k \overline{\mathbf{L}} \overline{\mathbf{B}}^k = \overline{\mathbf{L}}^3$, $\overline{\mathbf{W}} = \overline{\mathbf{L}}^2 \succ 0$ and hence (14) is satisfied for $\theta = \lambda_{\max}(\overline{\mathbf{L}})$. So, (15) says that if we choose $\gamma = 1/\theta$, then

$$\frac{1}{K} \sum_{k=0}^{K-1} \left\| \nabla f(x^k) \right\|_{\mathbf{I}}^2 \leq \frac{2\lambda_{\max}(\overline{\mathbf{L}})\left( f(x^0) - f(x^K) \right)}{K},$$

which exactly matches the rate of gradient descent in the non-convex setting. As for convergence of the iterates, the rate in (16) is $\lambda_{\max}(\overline{\mathbf{L}})/\lambda_{\min}(\overline{\mathbf{L}})$ which corresponds to the precise gradient descent result for strongly convex functions.

**2. Permutation.** Assume[2] $n = d$ and the use of `Perm-1` (special case of Definition 2) sketch $\mathbf{C}_i^k = n e_{\pi_i^k} e_{\pi_i^k}^\top$, where $\pi^k = (\pi_1^k, \ldots, \pi_n^k)$ is a random permutation of $[n]$. Then

$$\mathbb{E}\left[ \overline{\mathbf{B}}^k \right] = \frac{1}{n} \sum_{i=1}^n \mathbb{E}\left[ \mathbf{C}_i^k \mathbf{L}_i \mathbf{C}_i^k \right] = \frac{1}{n} \sum_{i=1}^n n\mathrm{Diag}(\mathbf{L}_i) = \sum_{i=1}^n \mathbf{D}_i = n \overline{\mathbf{D}},$$

---

[2]This is mainly done to simplify the presentation. Results can be generalized to the case of $n \neq d$ in a similar manner as in (Szlendak et al., 2022), which can be found in the Appendix.

where $\overline{\mathbf{D}} := \frac{1}{n} \sum_{i=1}^{n} \mathbf{D}_i, \mathbf{D}_i := \mathrm{Diag}(\mathbf{L}_i)$. Then inequality (14) leads to

$$n\,\overline{\mathbf{D}}\,\overline{\mathbf{L}}\,\overline{\mathbf{D}} \preceq \tfrac{\theta}{2}\left(\overline{\mathbf{L}}\,\overline{\mathbf{D}} + \overline{\mathbf{D}}\,\overline{\mathbf{L}}\right), \tag{17}$$

which may not always hold as $\overline{\mathbf{L}}\,\overline{\mathbf{D}} + \overline{\mathbf{D}}\,\overline{\mathbf{L}}$ is not guaranteed to be positive-definite—even in the case of $\overline{\mathbf{L}} \succ 0$. However, such a condition can be enforced via a slight modification of the permutation sketches, which is done in Section 3.2. The limitation of such an approach is that the resulting compressors are no longer unbiased.

**Remark 1.** *Matrix $\overline{\mathbf{W}}$ in the case of permutation sketches may not be positive-definite. Consider the following example of a homogeneous ($\mathbf{L}_i \equiv \mathbf{L}$) two-dimensional problem:*

$$\mathbf{L} = \begin{bmatrix} a & c \\ c & b \end{bmatrix}. \tag{18}$$

*Then*

$$\overline{\mathbf{W}} = \tfrac{1}{2}\left[\overline{\mathbf{L}}\,\overline{\mathbf{D}} + \overline{\mathbf{D}}\,\overline{\mathbf{L}}\right] = \begin{bmatrix} a^2 & c(a+b)/2 \\ c(a+b)/2 & b^2 \end{bmatrix}, \tag{19}$$

*which for $c > \frac{2ab}{a+b}$ has $\det(\overline{\mathbf{W}}) < 0$, and thus $\overline{\mathbf{W}} \nsucc 0$ according to Sylvester's criterion.*

Next, we focus on the particular case of **permutation** sketches, which are the most suitable for model partitioning according to Independent Subnetwork Training (IST). In the rest of this section, we discuss how the condition (14) can be enforced via a specially designed preconditioning of the problem (12) or modification of the sketch mechanism (7).

## 3.1 Homogeneous problem preconditioning

To start, consider a homogeneous setting $f_i(x) = \frac{1}{2}x^\top \mathbf{L}x$, so $\mathbf{L}_i \equiv \mathbf{L}$. Now define $\mathbf{D} = \mathrm{Diag}(\mathbf{L})$ – a diagonal matrix with elements equal to the diagonal of $\mathbf{L}$. Then, the problem can be converted to

$$f_i(\mathbf{D}^{-\frac{1}{2}}x) = \tfrac{1}{2}\left(\mathbf{D}^{-\frac{1}{2}}x\right)^\top \mathbf{L}\left(\mathbf{D}^{-\frac{1}{2}}x\right) = \tfrac{1}{2}x^\top\left(\mathbf{D}^{-\frac{1}{2}}\mathbf{L}\mathbf{D}^{-\frac{1}{2}}\right)x = \tfrac{1}{2}x^\top \tilde{\mathbf{L}}\,x \tag{20}$$

which is equivalent to the original problem after changing the variables $\tilde{x} := \mathbf{D}^{-\frac{1}{2}}x$. Note that $\mathbf{D} = \mathrm{Diag}(\mathbf{L})$ is positive-definite as $\mathbf{L} \succ 0$, and therefore $\tilde{\mathbf{L}} \succ 0$. Moreover, the preconditioned matrix $\tilde{\mathbf{L}}$ has all ones on the diagonal: $\mathrm{Diag}(\tilde{\mathbf{L}}) = \mathbf{I}$. If we now combine (20) with `Perm-1` sketches

$$\mathbb{E}\left[\overline{\mathbf{B}}^k\right] = \mathbb{E}\left[\tfrac{1}{n}\sum_{i=1}^{n}\mathbf{C}_i\tilde{\mathbf{L}}\mathbf{C}_i\right] = n\mathrm{Diag}(\tilde{\mathbf{L}}) = n\mathbf{I}.$$

Therefore, inequality (14) takes the form $\tilde{\mathbf{W}} = n\tilde{\mathbf{L}} \succeq \frac{1}{\theta}n^2\tilde{\mathbf{L}}$, which holds for $\theta \geq n$, and the left-hand side of (15) can be transformed (for an accurate comparison to standard methods) in the following way:

$$\left\|\nabla f(x^k)\right\|^2_{\tilde{\mathbf{L}}^{-1}\tilde{\mathbf{W}}\tilde{\mathbf{L}}^{-1}} \geq n\lambda_{\min}\left(\tilde{\mathbf{L}}^{-1}\right)\left\|\nabla f(x^k)\right\|^2_{\mathbf{I}} = n\lambda_{\max}(\tilde{\mathbf{L}})\left\|\nabla f(x^k)\right\|^2_{\mathbf{I}} \tag{21}$$

The resulting convergence guarantee is

$$\tfrac{1}{K}\sum_{k=0}^{K-1}\mathbb{E}\left[\left\|\nabla f(x^k)\right\|^2_{\mathbf{I}}\right] \leq \tfrac{2\lambda_{\max}(\tilde{\mathbf{L}})\left(f(x^0) - \mathbb{E}\left[f(x^K)\right]\right)}{K}, \tag{22}$$

which matches classical gradient descent.

## 3.2 Heterogeneous sketch preconditioning

In contrast to the homogeneous case, the heterogeneous problem $f_i(x) = \frac{1}{2}x^\top \mathbf{L}_i x$ cannot be so easily preconditioned by a simple change of variables $\tilde{x} := \mathbf{D}^{-\frac{1}{2}}x$, as every client $i$ has its own matrix $\mathbf{L}_i$. However, this problem can be fixed via the following modification of `Perm-1`, which scales the output according to the diagonal elements of the local smoothness matrix $\mathbf{L}_i$:

$$\tilde{\mathbf{C}}_i := \sqrt{n/\left[\mathbf{L}_i\right]_{\pi_i,\pi_i}}\,e_{\pi_i}e_{\pi_i}^\top. \tag{23}$$

In this case, $\mathbb{E}\left[\tilde{\mathbf{C}}_i \mathbf{L}_i \tilde{\mathbf{C}}_i\right] = \mathbf{I}$, $\mathbb{E}\left[\overline{\mathbf{B}}^k\right] = \mathbf{I}$, and $\overline{\mathbf{W}} = \overline{\mathbf{L}}$. Then inequality (14) is satisfied for $\theta \geq 1$.

If one inputs these results into (15), such convergence guarantee can be obtained

$$\frac{1}{K}\sum_{k=0}^{K-1}\mathbb{E}\left[\left\|\nabla f(x^k)\right\|_{\mathbf{I}}^2\right] \leq \frac{2\lambda_{\max}(\overline{\mathbf{L}})\left(f(x^0)-\mathbb{E}\left[f(x^K)\right]\right)}{K}, \tag{24}$$

which matches the gradient descent result as well. Thus, we can conclude that heterogeneity does not bring such a fundamental challenge in this scenario. In addition, the method with `Perm-1` is significantly better in terms of computational and communication complexity, as it requires calculation of the local gradients with respect to much smaller submodels and transmits only sparse updates.

This construction also shows that for $\gamma = 1/\theta = 1$

$$\gamma\lambda_{\min}\left(\overline{\mathbf{L}}^{-\frac{1}{2}}\overline{\mathbf{W}}\,\overline{\mathbf{L}}^{-\frac{1}{2}}\right) = \lambda_{\min}\left(\overline{\mathbf{L}}^{-\frac{1}{2}}\overline{\mathbf{L}}\,\overline{\mathbf{L}}^{-\frac{1}{2}}\right) = 1, \tag{25}$$

which, after plugging into the bound for the iterates (16), shows that the method basically converges in one iteration. This observation indicates that sketch preconditioning can be extremely efficient, although it uses only the diagonal elements of matrices $\mathbf{L}_i$.

Now that we understand that the method can perform very well in the special case of $\tilde{b}_i \equiv 0$, we can move on to a more complicated situation.

## 4 IRREDUCIBLE BIAS IN THE GENERAL CASE

Now we look at the most general heterogeneous case with different matrices and linear terms $f_i(x) \equiv \frac{1}{2}x^\top \mathbf{L}_i x - x^\top b_i$. In this instance, the gradient estimator (2) takes the form

$$g^k = \frac{1}{n}\sum_{i=1}^{n}\mathbf{C}_i^k\nabla f_i(\mathbf{C}_i^k x^k) = \frac{1}{n}\sum_{i=1}^{n}\mathbf{C}_i^k\left(\mathbf{L}_i\mathbf{C}_i^k x^k - b_i\right) = \overline{\mathbf{B}}^k x^k - \overline{\mathbf{Cb}}, \tag{26}$$

where $\overline{\mathbf{Cb}} = \frac{1}{n}\sum_{i=1}^{n}\mathbf{C}_i^k b_i$. Herewith let us use a heterogeneous permutation sketch preconditioner (23), as in Section 3.2. Then $\mathbb{E}\left[\overline{\mathbf{B}}^k\right] = \mathbf{I}$ and $\mathbb{E}\left[\overline{\mathbf{Cb}}\right] = \frac{1}{\sqrt{n}}\widetilde{\mathbf{D}\,\mathbf{b}}$, where $\widetilde{\mathbf{D}\,\mathbf{b}} := \frac{1}{n}\sum_{i=1}^{n}\mathbf{D}_i^{-\frac{1}{2}}b_i$. Furthermore, the expected gradient estimator (26) results in $\mathbb{E}\left[g^k\right] = x^k - \frac{1}{\sqrt{n}}\widetilde{\mathbf{D}\,\mathbf{b}}$ and can be transformed in the following manner:

$$\mathbb{E}\left[g^k\right] = \overline{\mathbf{L}}^{-1}\overline{\mathbf{L}}x^k \pm \overline{\mathbf{L}}^{-1}\overline{b} - \frac{1}{\sqrt{n}}\widetilde{\mathbf{D}\,\mathbf{b}} = \overline{\mathbf{L}}^{-1}\nabla f(x^k) + \underbrace{\overline{\mathbf{L}}^{-1}\overline{b} - \frac{1}{\sqrt{n}}\widetilde{\mathbf{D}\,\mathbf{b}}}_{h}, \tag{27}$$

which reflects the decomposition of the estimator into the optimally preconditioned true gradient and a bias, depending on the linear terms $b_i$.

### 4.1 BIAS OF THE METHOD

Estimator (27) can be directly plugged (with proper conditioning) into the general SGD update (2)

$$\mathbb{E}\left[x^{k+1}\right] = x^k - \gamma\mathbb{E}\left[g^k\right] = (1-\gamma)x^k + \frac{\gamma}{\sqrt{n}}\widetilde{\mathbf{D}\,\mathbf{b}} = (1-\gamma)^{k+1}x^0 + \frac{\gamma}{\sqrt{n}}\widetilde{\mathbf{D}\,\mathbf{b}}\sum_{j=0}^{k}(1-\gamma)^j. \tag{28}$$

The resulting recursion (28) is exact, and its asymptotic limit can be analyzed. Thus, for constant $\gamma < 1$, by using the formula for the sum of the first $k$ terms of a geometric series, one gets

$$\mathbb{E}\left[x^k\right] = (1-\gamma)^k x^0 + \frac{1-(1-\gamma)^k}{\sqrt{n}}\widetilde{\mathbf{D}\,\mathbf{b}} \xrightarrow[k\to\infty]{} \frac{1}{\sqrt{n}}\widetilde{\mathbf{D}\,\mathbf{b}},$$

which shows that in the limit, the first initialization term (with $x^0$) vanishes while the second converges to $\frac{1}{\sqrt{n}}\widetilde{\mathbf{D}\,\mathbf{b}}$. This reasoning shows that the method does not converge to the exact solution

$$\mathbb{E}\left[x^k\right] \to x^\infty \neq x^\star \in \arg\min_{x\in\mathbb{R}^d}\left\{\frac{1}{2}x^\top\overline{\mathbf{L}}x - x^\top\overline{b}\right\},$$

which for the positive-definite $\overline{\mathbf{L}}$ can be defined as $x^\star = \overline{\mathbf{L}}^{-1}\overline{b}$, while $x^\infty = \frac{1}{n\sqrt{n}}\sum_{i=1}^{n}\mathbf{D}_i^{-\frac{1}{2}}b_i$. So, in general, there is an unavoidable bias. However, in the limit case: $n = d \to \infty$, the bias diminishes.

## 4.2 GENERIC CONVERGENCE ANALYSIS

While the analysis in Section 4.1 is precise, it does not allow us to compare the convergence of IST to standard optimization methods. Therefore, we also analyze the non-asymptotic behavior of the method to understand the convergence speed. Our result is formalized in the following theorem:

**Theorem 2.** *Consider the method* (2) *with the estimator* (26) *for the quadratic problem* (12) *with the positive-definite matrix* $\overline{\mathbf{L}} \succ 0$*. Assume that for every* $\mathbf{D}_i := \mathrm{Diag}(\mathbf{L}_i)$ *matrices* $\mathbf{D}_i^{-\frac{1}{2}}$ *exist, scaled permutation sketches* (23) *are used, and heterogeneity is bounded as* $\mathbb{E}\left[\left\|g^k - \mathbb{E}\left[g^k\right]\right\|_{\overline{\mathbf{L}}}^2\right] \le \sigma^2$*. Then, for the step size chosen as follows:*

$$0 < \gamma \le \gamma_{c,\beta} := \frac{1/2 - \beta}{\beta + 1/2}, \tag{29}$$

*where* $\gamma_{c,\beta} \in (0, 1]$ *for* $\beta \in (0, 1/2)$*, the iterates satisfy*

$$\frac{1}{K}\sum_{k=0}^{K-1}\mathbb{E}\left[\left\|\nabla f(x^k)\right\|_{\overline{\mathbf{L}}^{-1}}^2\right] \le \frac{2\left(f(x^0) - \mathbb{E}\left[f(x^K)\right]\right)}{\gamma K} + \left(2\beta^{-1}\left(1 - \gamma\right) + \gamma\right)\|h\|_{\overline{\mathbf{L}}}^2 + \gamma\sigma^2, \tag{30}$$

*where* $\overline{\mathbf{L}} = \frac{1}{n}\sum_{i=1}^{n}\mathbf{L}_i, h = \overline{\mathbf{L}}^{-1}\overline{\mathrm{b}} - \frac{1}{n^{3/2}}\sum_{i=1}^{n}\mathbf{D}_i^{-\frac{1}{2}}\mathrm{b}_i$ *and* $\overline{\mathrm{b}} = \frac{1}{n}\sum_{i=1}^{n}\mathrm{b}_i$*.*

Note that the derived convergence upper bound has a neighborhood proportional to the bias of the gradient estimator $h$ and level of heterogeneity $\sigma^2$. Some of these terms with factor $\gamma$ can be eliminated by decreasing the learning rate (e.g., $\sim 1/\sqrt{k}$). However, such a strategy does not diminish the term with a multiplier $2\beta^{-1}(1 - \gamma)$, making the neighborhood irreducible. Moreover, this term can be eliminated for $\gamma = 1$, which also minimizes the first term that decreases as $1/K$. However, this step size choice maximizes the terms with factor $\gamma$. Thus, there exists an inherent trade-off between convergence speed and the size of the neighborhood.

In addition, convergence to the stationary point is measured by the weighted $\overline{\mathbf{L}}^{-1}$ squared norm of the gradient. At the same time, the neighborhood term depends on the weighted by $\overline{\mathbf{L}}$ norm of $h$. This fine-grained decoupling is achieved by carefully applying the Fenchel-Young inequality and provides a tighter characterization of the convergence compared to using standard Euclidean distances.

**Homogeneous case.** In this scenario, every worker has access to all data $f_i(x) \equiv \frac{1}{2}x^\top \mathbf{L}x - x^\top \mathrm{b}$. Then diagonal preconditioning of the problem can be used, as in the previous Section 3.1. This results in a gradient $\nabla f(x) = \tilde{\mathbf{L}}x - \tilde{\mathrm{b}}$ for $\tilde{\mathbf{L}} = \mathbf{D}^{-\frac{1}{2}}\mathbf{L}\mathbf{D}^{-\frac{1}{2}}$ and $\tilde{\mathrm{b}} = \mathbf{D}^{-\frac{1}{2}}\mathrm{b}$. If this expression is further combined with a permutation sketch scaled by $1/\sqrt{n}$ $\mathbf{C}_i' := \sqrt{n}e_{\pi_i}e_{\pi_i}^\top$, the resulting gradient estimator is:

$$g^k = x^k - \frac{1}{\sqrt{n}}\tilde{\mathrm{b}} = \tilde{\mathbf{L}}^{-1}\nabla f(x^k) + \tilde{h}, \tag{31}$$

for $\tilde{h} = \tilde{\mathbf{L}}^{-1}\tilde{\mathrm{b}} - \frac{1}{\sqrt{n}}\tilde{\mathrm{b}}$. In this case, the heterogeneity term $\sigma^2$ from the upper bound (30) disappears as $\mathbb{E}\left[\left\|g^k - \mathbb{E}\left[g^k\right]\right\|_{\overline{\mathbf{L}}}^2\right] = 0$, which significantly decreases the neighborhood size. However, the bias term depending on $\tilde{h}$ still remains, as the method does not converge to the exact solution $x^k \to x^\infty \ne x^\star = \tilde{\mathbf{L}}^{-1}\tilde{\mathrm{b}}$ for positive-definite $\tilde{\mathbf{L}}$. Nevertheless the method's fixed point $x^\infty = \tilde{\mathrm{b}}/\sqrt{n}$ and solution $x^\star$ can coincide when $\tilde{\mathbf{L}}^{-1}\tilde{\mathrm{b}} = \frac{1}{\sqrt{n}}\tilde{\mathrm{b}}$, which means that $\tilde{\mathrm{b}}$ is the right eigenvector of matrix $\tilde{\mathbf{L}}^{-1}$ with eigenvalue $\frac{1}{\sqrt{n}}$.

Let us contrast the obtained result (30) with the non-convex rate of SGD (Khaled & Richtárik, 2023) with constant step size $\gamma$ for $L$-smooth and lower-bounded $f$

$$\min_{k \in \{0,\dots,K-1\}}\left\|\nabla f(x^k)\right\|^2 \le \frac{6\left(f(x^0) - \inf f\right)}{\gamma K} + \gamma LC, \tag{32}$$

where constant $C$ depends, for example, on the variance of the stochastic gradient estimator. Observe that the first term in the compared upper bounds (32) and (30) is almost identical and decreases with speed $1/K$. However, unlike (30), the neighborhood for SGD can be completely eliminated by reducing the step size $\gamma$. This highlights a fundamental difference between our results and unbiased

methods. The intuition behind this issue is that for SGD-type methods like compressed gradient descent

$$x^{k+1} = x^k - \mathcal{C}(\nabla f(x^k)) \tag{33}$$

the gradient estimate is unbiased and enjoys the property that variance

$$\mathbb{E}\left[\|\mathcal{C}(\nabla f(x^k)) - \nabla f(x^k)\|^2\right] \leq \omega \|\nabla f(x^k)\|^2 \tag{34}$$

goes down to zero as the method progresses because $\nabla f(x^k) \to \nabla f(x^\star) = 0$ in the unconstrained case. In addition, any stationary point $x^\star$ ceases to be a fixed point of the iterative procedure as

$$x^\star \neq x^\star - \nabla f(\mathcal{C}(x^\star)), \tag{35}$$

in the general case, unlike for compressed gradient descent with both biased and unbiased compressors $\mathcal{C}$. Thus, even if the method—computing the gradient with a sparse model—is initialized from the *solution* after one gradient step, the method may get away from the optimum.

### 4.3 COMPARISON TO PREVIOUS WORKS

**Independent Subnetwork Training (Yuan et al., 2022).** There are several improvements over the previous works that tried to theoretically analyze the convergence of distributed IST. The first difference is that our results allow for an almost arbitrary level of model sparsification, i.e., will work for any $\omega \geq 0$ as permutation sketches can be viewed as a special case of compression operators (1). This represents a significant improvement over the work of Yuan et al. (2022), which demands[3] $\omega \lesssim \mu^2/L^2$. Such a requirement is very restrictive as the condition number $L/\mu$ of the loss function $f$ is typically very large for any non-trivial optimization problem. Thus, the sparsifier's (4) variance $\omega = d/q - 1$ has to be very close to 0 and $q \approx d$. Thus, the previous theory allows almost no compression (sparsification) because it is based on the analysis of gradient descent with compressed iterates (Khaled & Richtárik, 2019).

The second distinction is that the original IST work (Yuan et al., 2022) considered a single node setting, and thus their convergence bounds did not capture the effect of heterogeneity, which we believe is of crucial importance for distributed settings (Chraibi et al., 2019; Shulgin & Richtárik, 2022). Moreover, the original work considers the Lipschitz continuity of the loss function $f$, which is not satisfied for a simple quadratic model. A more detailed comparison, including additional assumptions on the gradient estimator made by Yuan et al. (2022), is presented in the Appendix D.

**FL with Model Pruning.** In a recent work, Zhou et al. (2022) made an attempt to analyze a variant of the FedAvg algorithm with sparse local initialization and compressed gradient training (pruned local models). They considered a case of $L$-smooth loss and a sparsification operator satisfying a similar condition to (1). However, they also assumed that the squared norm of the stochastic gradient is uniformly bounded (11), which is "pathological" (Khaled et al., 2020)—especially in the case of local methods—as it does not allow the analysis to capture the very important effect of heterogeneity and can result in vacuous bounds.

## 5 CONCLUSIONS AND FUTURE WORK

In this study, we introduced a novel approach to understanding training with combined model and data parallelism for a quadratic model. Our framework sheds light on distributed submodel optimization, which reveals the advantages and limitations of Independent Subnetwork Training (IST). Moreover, we accurately characterized the behavior of the considered method in both homogeneous and heterogeneous scenarios without imposing restrictive assumptions on the gradient estimators.

In future research, it would be valuable to explore extensions of our findings to settings that are closer to scenarios, such as cross-device federated learning. This could involve investigating partial participation support, leveraging local training benefits, and ensuring robustness against stragglers. Additionally, it would be interesting to generalize our results to non-quadratic scenarios without relying on pathological assumptions. Another potential promising research direction is algorithmic modifications of the original IST to solve the fundamental problems highlighted in this work and acceleration of training.

---

[3]$\mu$ refers to a constant from the Polyak-Łojasiewicz (or strong convexity) condition. In case of a quadratic problem with positive-definite matrix $\mathbf{A}$ constant $\mu$ equals to $\lambda_{\min}(\mathbf{A})$

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

# A  APPENDIX

## CONTENTS

# B  BASIC AND AUXILIARY FACTS

$\mathbf{L}$-matrix smoothness:

$$f(x + h) \le f(x) + \langle \nabla f(x), h \rangle + \frac{1}{2} \langle \mathbf{L}h, h \rangle, \qquad \forall x, h \in \mathbb{R}^d. \tag{36}$$

**Basic Inequalities.**  For all vectors $a, b \in \mathbb{R}^d$ and random vector $X \in \mathbb{R}^d$:

$$2\langle a, b \rangle = \|a\|^2 + \|b\|^2 - \|a - b\|^2, \tag{37}$$

$$\mathbf{E} \|X - a\|^2 = \mathbf{E} \|X - \mathbf{E}\, X\|^2 + \|\mathbf{E}\, X - a\|^2. \tag{38}$$

**Lemma 1** (Fenchel–Young inequality). *For any function $f$ and its convex conjugate $f^*$, Fenchel's inequality (also known as the Fenchel–Young inequality) holds for every $x, y \in \mathbb{R}^d$*

$$\langle x, y \rangle \le f(x) + f^*(y).$$

*The proof follows from the definition of conjugate:* $f^*(y) := \sup_{x'} \{ \langle y, x' \rangle - f(x') \} \ge \langle y, x \rangle - f(x)$.

In the case of a quadratic function $f(x) = \beta \|x\|_{\mathbf{L}}^2$, we can compute $f^*(y) = \frac{1}{4} \beta^{-1} \|y\|_{\mathbf{L}^{-1}}^2$. Thus

$$\langle x, y \rangle \le \beta \|x\|_{\mathbf{L}}^2 + \frac{1}{4} \beta^{-1} \|y\|_{\mathbf{L}^{-1}}^2. \tag{39}$$

# C  PROOFS

## C.1  PERMUTATION SKETCH COMPUTATIONS

All derivations in this section are performed for the $n = d$ case.

**Classical Permutation Sketching.**  `Perm-1`: $\mathbf{C}_i = n e_{\pi_i} e_{\pi_i}^\top$, where $\pi = (\pi_1, \dots, \pi_n)$ is a random permutation of $[n]$. For the homogeneous problem $\mathbf{L}_i \equiv \mathbf{L}$:

$$\mathbb{E}\left[ \overline{\mathbf{B}}^k \right] = \mathbb{E}\left[ \frac{1}{n} \sum_{i=1}^n \mathbf{C}_i \, \mathbf{L} \, \mathbf{C}_i \right] = n \mathrm{Diag}(\mathbf{L}) \tag{40}$$

Then

$$2\,\overline{\mathbf{W}} = \mathbb{E}\left[ \mathbf{L}\, \overline{\mathbf{B}}^k + \overline{\mathbf{B}}^k \mathbf{L} \right] = n \left( \mathbf{L}\mathrm{Diag}(\mathbf{L}) + \mathrm{Diag}(\mathbf{L})\mathbf{L} \right) \tag{41}$$

and

$$\mathbb{E}\left[ \overline{\mathbf{B}}^k \mathbf{L}\, \overline{\mathbf{B}}^k \right] = n^2 \mathrm{Diag}(\mathbf{L})\mathbf{L}\mathrm{Diag}(\mathbf{L}). \tag{42}$$

By repeating basically the same calculations for $\mathbf{C}_i' = \sqrt{n} e_{\pi_i} e_{\pi_i}^\top$ we have that

$$\mathbb{E}\left[ \overline{\mathbf{B}}^k \right] = \mathbb{E}\left[ \frac{1}{n} \sum_{i=1}^n \mathbf{C}_i' \mathbf{L} \mathbf{C}_i' \right] = \mathrm{Diag}(\mathbf{L}), \tag{43}$$

and $\mathbb{E}\left[ \overline{\mathbf{B}}^k \mathbf{L}\, \overline{\mathbf{B}}^k \right] = \mathrm{Diag}(\mathbf{L})\mathbf{L}\mathrm{Diag}(\mathbf{L})$, $2\,\overline{\mathbf{W}} = \mathbb{E}\left[ \mathbf{L}\, \overline{\mathbf{B}}^k + \overline{\mathbf{B}}^k \mathbf{L} \right] = \mathbf{L}\mathrm{Diag}(\mathbf{L}) + \mathrm{Diag}(\mathbf{L})\mathbf{L}$.

### C.1.1  HETEROGENEOUS SKETCH PRECONDITIONING.

We recall the following modification of `Perm-1`:

$$\tilde{\mathbf{C}}_i := \sqrt{n / [\mathbf{L}_i]_{\pi_i, \pi_i}} \, e_{\pi_i} e_{\pi_i}^\top. \tag{44}$$

Then

$$\mathbb{E}\left[ \tilde{\mathbf{C}}_i \mathbf{L}_i \tilde{\mathbf{C}}_i \right] = \mathbb{E}\left[ n [\mathbf{L}_i]_{\pi_i, \pi_i}^{-1} e_{\pi_i} e_{\pi_i}^\top \mathbf{L}_i e_{\pi_i} e_{\pi_i}^\top \right] = \frac{1}{n} \sum_{j=1}^n n e_j \mathbf{I}_{j,j} e_j^\top = \mathbf{I}. \tag{45}$$

and

$$
\begin{aligned}
\mathbb{E}\left[\overline{\mathbf{B}}^k\right] &= \mathbb{E}\left[\frac{1}{n}\sum_{i=1}^n \tilde{\mathbf{C}}_i \mathbf{L}_i \tilde{\mathbf{C}}_i\right] \\
&= \frac{1}{n}\sum_{i=1}^n \mathbb{E}\left[n[\mathbf{L}_i]_{\pi_i,\pi_i}^{-1} e_{\pi_i} e_{\pi_i}^\top \mathbf{L}_i e_{\pi_i} e_{\pi_i}^\top\right] \\
&= \frac{1}{n}\sum_{i=1}^n \frac{1}{n}\sum_{j=1}^n n[\mathbf{L}_i]_{j,j}^{-1} e_j [\mathbf{L}_i]_{j,j} e_j^\top \\
&= \frac{1}{n}\sum_{i=1}^n \sum_{j=1}^n e_j e_j^\top \\
&= \mathbf{I}.
\end{aligned}
$$

Thus $\overline{\mathbf{W}} = \frac{1}{2}\mathbb{E}\left[\overline{\mathbf{L}}\,\overline{\mathbf{B}}^k + \overline{\mathbf{B}}^k\,\overline{\mathbf{L}}\right] = \overline{\mathbf{L}}$. On the left hand side of inequality (14), we have

$$
\begin{aligned}
\mathbb{E}\left[\overline{\mathbf{B}}^k\,\overline{\mathbf{L}}\,\overline{\mathbf{B}}^k\right] &= \mathbb{E}\left[\frac{1}{n}\sum_{i=1}^n \tilde{\mathbf{C}}_i \mathbf{L}_i \tilde{\mathbf{C}}_i \,\overline{\mathbf{L}}\, \frac{1}{n}\sum_{i=j}^n \tilde{\mathbf{C}}_j \mathbf{L}_j \tilde{\mathbf{C}}_j\right] \\
&= \frac{1}{n^2}\sum_{i,j=1}^n \mathbb{E}\left[\tilde{\mathbf{C}}_i \mathbf{L}_i \tilde{\mathbf{C}}_i \,\overline{\mathbf{L}}\, \tilde{\mathbf{C}}_j \mathbf{L}_j \tilde{\mathbf{C}}_j\right] \\
&= \sum_{i,j=1}^n e_i e_i^\top \,\overline{\mathbf{L}}\, e_j e_j^\top \\
&= \mathbf{I}\,\overline{\mathbf{L}}\,\mathbf{I} \\
&= \overline{\mathbf{L}}.
\end{aligned}
$$

## C.2 INTERPOLATION CASE: PROOF OF THEOREM 1

In the quadratic interpolation regime, the linear term is zero $f_i(x) = \frac{1}{2}x^\top \mathbf{L}_i x$, and the gradient estimator has the form

$$
g^k = \frac{1}{n}\sum_{i=1}^n \mathbf{C}_i^k \nabla f_i(\mathbf{C}_i^k x^k) = \frac{1}{n}\sum_{i=1}^n \mathbf{C}_i^k \mathbf{L}_i \mathbf{C}_i^k x^k = \overline{\mathbf{B}}^k x^k. \tag{46}
$$

*Proof.* First, we prove the **stationary point** convergence result (15).

Using the $\overline{\mathbf{L}}$-smoothness of function $f$, we get

$$
\begin{aligned}
f(x^{k+1}) \overset{(2)}{=} f(x^k - \gamma g^k) \quad &\overset{(8)}{\leq}\quad f(x^k) - \langle \nabla f(x^k), \gamma g^k \rangle + \frac{\gamma^2}{2}\|g^k\|_{\overline{\mathbf{L}}}^2 \\
&\overset{(13)}{=}\quad f(x^k) - \gamma\left\langle \overline{\mathbf{L}}\,x^k, \overline{\mathbf{B}}^k x^k \right\rangle + \frac{\gamma^2}{2}\left\|\overline{\mathbf{B}}^k x^k\right\|_{\overline{\mathbf{L}}}^2 \\
&=\quad f(x^k) - \gamma(x^k)^\top \overline{\mathbf{L}}\,\overline{\mathbf{B}}^k x^k + \frac{\gamma^2}{2}(x^k)^\top \overline{\mathbf{B}}^k \overline{\mathbf{L}}\,\overline{\mathbf{B}}^k x^k.
\end{aligned}
$$

After applying conditional expectation, using its linearity, and the fact that

$$
x^\top \mathbf{A} x = \frac{1}{2}x^\top \left(\mathbf{A} + \mathbf{A}^\top\right) x
$$

we get

$$
\begin{aligned}
\mathbb{E}\left[f(x^{k+1}) \mid x^k\right] &\leq f(x^k) - \gamma (x^k)^\top \mathbb{E}\left[\overline{\mathbf{L}}\,\overline{\mathbf{B}}^k\right] x^k + \frac{\gamma^2}{2}(x^k)^\top \mathbb{E}\left[\overline{\mathbf{B}}^k\,\overline{\mathbf{L}}\,\overline{\mathbf{B}}^k\right] x^k \\
&= f(x^k) - \gamma (x^k)^\top \overline{\mathbf{W}}\, x^k + \frac{\gamma^2}{2}(x^k)^\top \mathbb{E}\left[\overline{\mathbf{B}}^k\,\overline{\mathbf{L}}\,\overline{\mathbf{B}}^k\right] x^k \\
&= f(x^k) - \gamma (\nabla f(x^k))^\top \overline{\mathbf{L}}^{-1}\,\overline{\mathbf{W}}\,\overline{\mathbf{L}}^{-1}\,\nabla f(x^k) \\
&\quad + \frac{\gamma^2}{2}(\nabla f(x^k))^\top \overline{\mathbf{L}}^{-1}\,\mathbb{E}\left[\overline{\mathbf{B}}^k\,\overline{\mathbf{L}}\,\overline{\mathbf{B}}^k\right] \overline{\mathbf{L}}^{-1}\,\nabla f(x^k) \\
&\overset{(14)}{\leq} f(x^k) - \gamma\|\nabla f(x^k)\|^2_{\overline{\mathbf{L}}^{-1}\,\overline{\mathbf{W}}\,\overline{\mathbf{L}}^{-1}} + \frac{\theta\gamma^2}{2}\|\nabla f(x^k)\|^2_{\overline{\mathbf{L}}^{-1}\,\overline{\mathbf{W}}\,\overline{\mathbf{L}}^{-1}} \\
&= f(x^k) - \gamma\left(1 - \theta\gamma/2\right)\|\nabla f(x^k)\|^2_{\overline{\mathbf{L}}^{-1}\,\overline{\mathbf{W}}\,\overline{\mathbf{L}}^{-1}} \\
&\leq f(x^k) - \frac{\gamma}{2}\|\nabla f(x^k)\|^2_{\overline{\mathbf{L}}^{-1}\,\overline{\mathbf{W}}\,\overline{\mathbf{L}}^{-1}},
\end{aligned}
$$

where the last inequality holds for the stepsize $\gamma \leq \frac{1}{\theta}$.

Rearranging gives

$$
\left\|\nabla f(x^k)\right\|^2_{\overline{\mathbf{L}}^{-1}\,\overline{\mathbf{W}}\,\overline{\mathbf{L}}^{-1}} \leq \frac{2}{\gamma}\left(f(x^k) - \mathbb{E}\left[f(x^{k+1}) \mid x^k\right]\right),
$$

which after averaging gives the desired result

$$
\frac{1}{K}\sum_{k=0}^{K-1}\mathbb{E}\left[\left\|\nabla f(x^k)\right\|^2_{\overline{\mathbf{L}}^{-1}\,\overline{\mathbf{W}}\,\overline{\mathbf{L}}^{-1}}\right] \leq \frac{2}{\gamma K}\sum_{k=0}^{K-1}\left(f(x^k) - \mathbb{E}\left[f(x^{k+1})\right]\right) = \frac{2\left(f(x^0) - \mathbb{E}\left[f(x^K)\right]\right)}{\gamma K}. \tag{47}
$$

Now we show the result for the **iterates convergence** (16).

Expectation conditioned on $x^k$:

$$
\begin{aligned}
\mathbb{E}\left[\|x^{k+1} - x^\star\|^2_{\overline{\mathbf{L}}}\right] &= \mathbb{E}\left[\|x^k - \gamma g^k - x^\star\|^2_{\overline{\mathbf{L}}}\right] \\
&= \|x^k - x^\star\|^2_{\overline{\mathbf{L}}} - 2\gamma\left\langle x^k - x^\star, \mathbb{E}\left[\overline{\mathbf{L}}\,\overline{\mathbf{B}}^k\right](x^k - x^\star)\right\rangle \\
&\quad + \gamma^2\left\langle \mathbb{E}\left[\overline{\mathbf{B}}^k\,\overline{\mathbf{L}}\,\overline{\mathbf{B}}^k\right](x^k - x^\star), x^k - x^\star\right\rangle \\
&\overset{x^\star=0}{=} \|x^k - x^\star\|^2_{\overline{\mathbf{L}}} - 2\gamma\left\langle x^k - x^\star, \overline{\mathbf{W}}(x^k - x^\star)\right\rangle \\
&\quad + \gamma^2\left\langle x^k - x^\star, \mathbb{E}\left[\overline{\mathbf{B}}^k\,\overline{\mathbf{L}}\,\overline{\mathbf{B}}^k\right](x^k - x^\star)\right\rangle \\
&\overset{(16)}{\leq} \|x^k - x^\star\|^2_{\overline{\mathbf{L}}} - 2\gamma\left\langle x^k - x^\star, \overline{\mathbf{W}}(x^k - x^\star)\right\rangle + \theta\gamma^2\left\langle x^k - x^\star, \overline{\mathbf{W}}(x^k - x^\star)\right\rangle \\
&= \|x^k - x^\star\|^2_{\overline{\mathbf{L}}} - 2\gamma\left(1 - \theta\gamma/2\right)\left\|\overline{\mathbf{L}}^{\frac{1}{2}}(x^k - x^\star)\right\|^2_{\overline{\mathbf{L}}^{-\frac{1}{2}}\,\overline{\mathbf{W}}\,\overline{\mathbf{L}}^{-\frac{1}{2}}} \\
&\overset{\gamma \leq 1/\theta}{\leq} \|x^k - x^\star\|^2_{\overline{\mathbf{L}}} - \gamma\left\|\overline{\mathbf{L}}^{\frac{1}{2}}(x^k - x^\star)\right\|^2_{\overline{\mathbf{L}}^{-\frac{1}{2}}\,\overline{\mathbf{W}}\,\overline{\mathbf{L}}^{-\frac{1}{2}}} \\
&\leq \|x^k - x^\star\|^2_{\overline{\mathbf{L}}} - \gamma\lambda_{\min}\left(\overline{\mathbf{L}}^{-\frac{1}{2}}\,\overline{\mathbf{W}}\,\overline{\mathbf{L}}^{-\frac{1}{2}}\right)\left\|\overline{\mathbf{L}}^{\frac{1}{2}}(x^k - x^\star)\right\|^2 \\
&= \left(1 - \gamma\lambda_{\min}\left(\overline{\mathbf{L}}^{-\frac{1}{2}}\,\overline{\mathbf{W}}\,\overline{\mathbf{L}}^{-\frac{1}{2}}\right)\right)\|x^k - x^\star\|^2_{\overline{\mathbf{L}}}.
\end{aligned}
$$

After unrolling the recursion we obtain the convergence result

$$
\mathbb{E}\left[\|x^{k+1} - x^\star\|^2_{\overline{\mathbf{L}}}\right] \leq \left(1 - \gamma\lambda_{\min}\left(\overline{\mathbf{L}}^{-\frac{1}{2}}\,\overline{\mathbf{W}}\,\overline{\mathbf{L}}^{-\frac{1}{2}}\right)\right)^{k+1}\|x^0 - x^\star\|^2_{\overline{\mathbf{L}}}.
$$

$\square$

## C.3 NON-ZERO SOLUTION

As a reminder, in the most general case, the problem has the form

$$f(x) = \frac{1}{n}\sum_{i=1}^n f_i(x), \qquad f_i(x) \equiv \frac{1}{2}x^\top \mathbf{L}_i x - x^\top \mathbf{b}_i \,.$$

with the gradient estimator

$$g^k = \frac{1}{n}\sum_{i=1}^n \mathbf{C}_i^k \nabla f_i(\mathbf{C}_i^k x^k) = \frac{1}{n}\sum_{i=1}^n \mathbf{C}_i^k \left(\mathbf{L}_i \mathbf{C}_i^k x^k - \mathbf{b}_i\right) = \overline{\mathbf{B}}^k x^k - \frac{1}{n}\sum_{i=1}^n \mathbf{C}_i^k \mathbf{b}_i \,. \qquad (48)$$

**General calculations for estimator** (26). In the heterogeneous case, the following sketch preconditioner is used

$$\tilde{\mathbf{C}}_i := \sqrt{n/\left[\mathbf{L}_i\right]_{\pi_i,\pi_i}} e_{\pi_i} e_{\pi_i}^\top.$$

Then $\mathbb{E}\left[\overline{\mathbf{B}}^k\right] = \mathbf{I}$ (calculation was done as in Section C.1.1) and

$$
\begin{aligned}
\mathbb{E}\left[\overline{\mathbf{C}}\mathbf{b}\right] &= \frac{1}{n}\sum_{i=1}^n \mathbb{E}\left[\tilde{\mathbf{C}}_i^k \mathbf{b}_i\right] \\
&= \frac{1}{n}\sum_{i=1}^n \mathbb{E}\left[\sqrt{n}[\mathbf{L}_i]_{\pi_i,\pi_i}^{-\frac{1}{2}} e_{\pi_i} e_{\pi_i}^\top \mathbf{b}_i\right] \\
&= \frac{1}{n}\sum_{i=1}^n \frac{1}{n}\sum_{j=1}^n \sqrt{n}[\mathbf{L}_i]_{j,j}^{-\frac{1}{2}} e_j [\mathbf{b}_i]_j \\
&= \frac{1}{n}\sum_{i=1}^n \frac{1}{n}\sqrt{n}\mathbf{D}_i^{-\frac{1}{2}} \mathbf{b}_i \\
&= \frac{1}{\sqrt{n}}\frac{1}{n}\sum_{i=1}^n \mathbf{D}_i^{-\frac{1}{2}} \mathbf{b}_i \\
&= \frac{1}{\sqrt{n}}\underbrace{\overline{\mathbf{D}^{-\frac{1}{2}}\mathbf{b}}}_{\widetilde{\mathbf{D}\,\mathbf{b}}}
\end{aligned}
$$

### C.3.1 CONVERGENCE ANALYSIS FOR HETEROGENEOUS CASE: PROOF OF THEOREM 2.

Here we formulate and further prove a more general version of Theorem 2, which is obtained as a special case of the next result for $c = 1/2$.

**Theorem 3.** *Consider the method* (2) *with estimator* (26) *for a quadratic problem* (12) *with positive-definite matrix* $\overline{\mathbf{L}} \succ 0$. *Then, if for every* $\mathbf{D}_i := \mathrm{Diag}(\mathbf{L}_i)$ *matrices* $\mathbf{D}_i^{-\frac{1}{2}}$ *exist, scaled permutation sketches* $\mathbf{C}_i := \sqrt{n}[\mathbf{L}_i^{-\frac{1}{2}}]_{\pi_i,\pi_i} e_{\pi_i} e_{\pi_i}^\top$ *are used and heterogeneity is bounded as* $\mathbb{E}\left[\left\|g^k - \mathbb{E}\left[g^k\right]\right\|_{\overline{\mathbf{L}}}^2\right] \le \sigma^2$. *Then, the step size is chosen as*

$$0 < \gamma \le \gamma_{c,\beta} := \frac{1 - c - \beta}{\beta + 1/2}, \qquad (49)$$

*where* $\gamma_{c,\beta} \in (0,1]$ *for* $\beta + c < 1$, *the iterates satisfy*

$$\frac{1}{K}\sum_{k=0}^{K-1}\mathbb{E}\left[\left\|\nabla f(x^k)\right\|_{\overline{\mathbf{L}}^{-1}}^2\right] \le \frac{f(x^0) - \mathbb{E}\left[f(x^K)\right]}{c\gamma K} + \left(\frac{1-\gamma}{c\beta} + \frac{\gamma}{2c}\right)\|h\|_{\overline{\mathbf{L}}}^2 + \frac{\gamma}{2c}\sigma^2. \qquad (50)$$

*where* $\overline{\mathbf{L}} = \frac{1}{n}\sum_{i=1}^n \mathbf{L}_i, h = \overline{\mathbf{L}}^{-1}\overline{\mathbf{b}} - \frac{1}{\sqrt{n}}\frac{1}{n}\sum_{i=1}^n \mathbf{D}_i^{-\frac{1}{2}} \mathbf{b}_i$ *and* $\overline{\mathbf{b}} = \frac{1}{n}\sum_{i=1}^n \mathbf{b}_i$.

*Proof.* By using $\mathbf{L}$-smoothness

$$\mathbb{E}\left[f(x^{k+1}) \mid x^k\right] \stackrel{(8)}{\leq} f(x^k) - \gamma\left\langle \nabla f(x^k), \mathbb{E}\left[g^k\right]\right\rangle + \frac{\gamma^2}{2}\mathbb{E}\left[\|g^k\|_{\mathbf{L}}^2\right]$$

$$\stackrel{(27),(38)}{=} f(x^k) - \gamma\left\langle \nabla f(x^k), \overline{\mathbf{L}}^{-1}\nabla f(x^k) + h\right\rangle$$

$$+ \frac{\gamma^2}{2}\left(\|\mathbb{E}\left[g^k\right]\|_{\overline{\mathbf{L}}}^2 + \mathbb{E}\left[\|g^k - \mathbb{E}\left[g^k\right]\|_{\overline{\mathbf{L}}}^2\right]\right)$$

$$\stackrel{(27)}{=} f(x^k) - \gamma\left(\left\langle \nabla f(x^k), \overline{\mathbf{L}}^{-1}\nabla f(x^k)\right\rangle + \left\langle \nabla f(x^k), h\right\rangle\right)$$

$$+ \frac{\gamma^2}{2}\left(\left\|\overline{\mathbf{L}}^{-1}\nabla f(x^k) + h\right\|_{\overline{\mathbf{L}}}^2 + \mathbb{E}\left[\|g^k - \mathbb{E}\left[g^k\right]\|_{\overline{\mathbf{L}}}^2\right]\right)$$

$$\stackrel{(37)}{=} f(x^k) - \gamma\left(\|\nabla f(x^k)\|_{\overline{\mathbf{L}}^{-1}}^2 + \left\langle \nabla f(x^k), h\right\rangle\right) + \frac{\gamma^2}{2}\mathbb{E}\left[\|g^k - \mathbb{E}\left[g^k\right]\|_{\overline{\mathbf{L}}}^2\right]$$

$$+ \frac{\gamma^2}{2}\left(\|\nabla f(x^k)\|_{\overline{\mathbf{L}}^{-1}}^2 + 2\left\langle \nabla f(x^k), h\right\rangle + \|h\|_{\overline{\mathbf{L}}}^2\right)$$

$$\leq f(x^k) - \gamma\left(1 - \gamma/2\right)\|\nabla f(x^k)\|_{\overline{\mathbf{L}}^{-1}}^2 + \frac{\gamma^2}{2}\sigma^2$$

$$- \gamma\left(1 - \gamma\right)\left\langle \nabla f(x^k), h\right\rangle + \frac{\gamma^2}{2}\|h\|_{\overline{\mathbf{L}}}^2,$$

where the last inequality follows from the grouping of similar terms and bounded heterogeneity

$$\mathbb{E}\left[\|g^k - \mathbb{E}\left[g^k\right]\|_{\overline{\mathbf{L}}}^2\right] = \mathbb{E}\left[\left\|g^k - \left(\overline{\mathbf{L}}^{-1}\nabla f(x^k) + h\right)\right\|_{\overline{\mathbf{L}}}^2\right] \tag{51}$$

$$= \mathbb{E}\left[\left\|\overline{\mathbf{B}}^k x^k - \overline{\mathbf{Cb}} - \left(x^k - \frac{1}{\sqrt{n}}\widetilde{\mathbf{Db}}\right)\right\|_{\overline{\mathbf{L}}}^2\right] \leq \sigma^2. \tag{52}$$

Next, using a Fenchel-Young inequality (39) for $\left\langle \nabla f(x^k), -h\right\rangle$ and $1 - \gamma \geq 0$

$$\mathbb{E}\left[f(x^{k+1}) \mid x^k\right] \leq f(x^k) - \gamma\left(1 - \gamma/2\right)\|\nabla f(x^k)\|_{\overline{\mathbf{L}}^{-1}}^2 + \frac{\gamma^2}{2}\left(\|h\|_{\overline{\mathbf{L}}}^2 + \sigma^2\right)$$

$$+ \gamma\left(1 - \gamma\right)\left[\beta\|\nabla f(x^k)\|_{\overline{\mathbf{L}}^{-1}}^2 + 0.25\beta^{-1}\|h\|_{\overline{\mathbf{L}}}^2\right]$$

$$\leq f(x^k) - \gamma\left(1 - \gamma/2 - \beta\left(1 - \gamma\right)\right)\|\nabla f(x^k)\|_{\overline{\mathbf{L}}^{-1}}^2$$

$$+ \gamma\left\{\left(\beta^{-1}\left(1 - \gamma\right) + \frac{\gamma}{2}\right)\|h\|_{\overline{\mathbf{L}}}^2 + \frac{\gamma}{2}\sigma^2\right\}, \tag{53}$$

where in the last inequality we grouped similar terms and used the fact that $0.25 < 1$.

Now to guarantee that $1 - \gamma/2 - \beta(1 - \gamma) \geq c > 0$, we choose the step size using

$$0 < \gamma \leq \gamma_{c,\beta} := \frac{1 - c - \beta}{\beta + 1/2}, \tag{54}$$

where $\gamma_{c,\beta} > 0$ for $\beta + c < 1$. This means that $\beta$ can not arbitrarily grow to diminish $\beta^{-1}$. Then, after standard manipulations and unrolling the recursion

$$\gamma c\|\nabla f(x^k)\|_{\overline{\mathbf{L}}^{-1}}^2 \leq f(x^k) - \mathbb{E}\left[f(x^{k+1}) \mid x^k\right] + \gamma\left(\beta^{-1}\left(1 - \gamma\right) + \gamma/2\right)\|h\|_{\overline{\mathbf{L}}}^2 + \frac{\gamma^2}{2}\sigma^2 \tag{55}$$

we obtain

$$\frac{c}{K}\sum_{k=0}^{K-1}\mathbb{E}\left[\|\nabla f(x^k)\|_{\overline{\mathbf{L}}^{-1}}^2\right] \leq \frac{f(x^0) - \mathbb{E}\left[f(x^K)\right]}{\gamma K} + \left(\beta^{-1}\left(1 - \gamma\right) + \gamma/2\right)\|h\|_{\overline{\mathbf{L}}}^2 + \frac{\gamma}{2}\sigma^2. \tag{56}$$

$\square$

### C.3.2 Homogeneous case

The main difference compared to the result in the previous subsection is that the gradient estimator expression (31) holds deterministically (without expectation $\mathbb{E}$). That is why $g^k = \mathbb{E}\left[g^k\right]$ and heterogeneity term $\sigma^2$ equals to 0.

We provide the full statement and proof for the homogeneous result discussed in 4.2.

**Theorem 4.** *Consider the method* (2) *with estimator* (31) *for a homogeneous quadratic problem* (12) *with positive-definite matrix* $\mathbf{L}_i \equiv \mathbf{L} \succ 0$. *Then if exists* $\mathbf{D}^{-\frac{1}{2}}$ *for* $\mathbf{D} := \mathrm{Diag}(\mathbf{L})$, *scaled permutation sketch* $\mathbf{C}'_i = \sqrt{n}e_{\pi_i}e_{\pi_i}^\top$ *is used and the step size is chosen as*

$$0 < \gamma \le \gamma_{c,\beta} := \frac{1 - c - \beta}{\beta + 1/2}, \tag{57}$$

*where* $\gamma_{c,\beta} > 0$ *for* $\beta + c < 1$. *Then the iterates satisfy*

$$\frac{1}{K}\sum_{k=0}^{K-1}\mathbb{E}\left[\left\|\nabla f(x^k)\right\|_{\tilde{\mathbf{L}}^{-1}}^2\right] \le \frac{f(x^0) - \mathbb{E}\left[f(x^K)\right]}{c\gamma K} + \left(\frac{1-\gamma}{c\beta} + \frac{\gamma}{2c}\right)\|h\|_{\tilde{\mathbf{L}}}^2, \tag{58}$$

*where* $\tilde{\mathbf{L}} = \mathbf{D}^{-\frac{1}{2}}\mathbf{L}\mathbf{D}^{-\frac{1}{2}}, h = \tilde{\mathbf{L}}^{-1}\tilde{\mathrm{b}} - \frac{1}{\sqrt{n}}\tilde{\mathrm{b}}$ *and* $\tilde{\mathrm{b}} = \mathbf{D}^{-\frac{1}{2}}\mathrm{b}$.

*Proof.* By using $\mathbf{L}$-smoothness

$$
\begin{aligned}
\mathbb{E}\left[f(x^k - \gamma g^k) \mid x^k\right] &\overset{(8)}{\le} f(x^k) - \left\langle \nabla f(x^k), \gamma\mathbb{E}\left[g^k\right]\right\rangle + \frac{\gamma^2}{2}\mathbb{E}\left[\left\|g^k\right\|_{\mathbf{L}}^2\right] \\
&\le f(x^k) - \gamma\left\langle \nabla f(x^k), \tilde{\mathbf{L}}^{-1}\nabla f(x^k) + h\right\rangle + \frac{\gamma^2}{2}\left\|\tilde{\mathbf{L}}^{-1}\nabla f(x^k) + h\right\|_{\tilde{\mathbf{L}}}^2 \\
&\overset{(37)}{=} f(x^k) - \gamma\left(\left\langle \nabla f(x^k), \tilde{\mathbf{L}}^{-1}\nabla f(x^k)\right\rangle + \left\langle \nabla f(x^k), h\right\rangle\right) \\
&\quad + \frac{\gamma^2}{2}\left(\left\|\nabla f(x^k)\right\|_{\tilde{\mathbf{L}}^{-1}}^2 + 2\left\langle \nabla f(x^k), h\right\rangle + \|h\|_{\tilde{\mathbf{L}}}^2\right) \\
&= f(x^k) - \gamma\left(1 - \gamma/2\right)\left\|\nabla f(x^k)\right\|_{\tilde{\mathbf{L}}^{-1}}^2 + \frac{\gamma^2}{2}\|h\|_{\tilde{\mathbf{L}}}^2 - \gamma\left(1 - \gamma\right)\left\langle \nabla f(x^k), h\right\rangle
\end{aligned}
$$

Next by using a Fenchel-Young inequality (39) for $\left\langle \nabla f(x^k), -h\right\rangle$ and $1 - \gamma \ge 0$

$$
\begin{aligned}
\mathbb{E}\left[f(x^{k+1}) \mid x^k\right] &\le f(x^k) - \gamma\left(1 - \gamma/2\right)\left\|\nabla f(x^k)\right\|_{\tilde{\mathbf{L}}^{-1}}^2 + \frac{\gamma^2}{2}\|h\|_{\tilde{\mathbf{L}}}^2 \\
&\quad + \gamma\left(1 - \gamma\right)\left[\beta\|\nabla f(x^k)\|_{\tilde{\mathbf{L}}^{-1}}^2 + 0.25\beta^{-1}\|h\|_{\tilde{\mathbf{L}}}^2\right] \\
&= f(x^k) - \gamma\left(1 - \gamma/2 - \beta(1 - \gamma)\right)\left\|\nabla f(x^k)\right\|_{\tilde{\mathbf{L}}^{-1}}^2 \\
&\quad + \gamma\left(\beta^{-1}\left(1 - \gamma\right) + \gamma/2\right)\|h\|_{\tilde{\mathbf{L}}}^2.
\end{aligned}
$$

Now to guarantee that $1 - \gamma/2 - \beta(1 - \gamma) \ge c > 0$ we choose the step size as

$$0 < \gamma \le \gamma_{c,\beta} := \frac{1 - c - \beta}{\beta + 1/2}, \tag{59}$$

where $\gamma_{c,\beta} \ge 0$ for $\beta + c < 1$.
Then after standard manipulations and unrolling the recursion

$$\gamma c\left\|\nabla f(x^k)\right\|_{\tilde{\mathbf{L}}^{-1}}^2 \le f(x^k) - \mathbb{E}\left[f(x^{k+1}) \mid x^k\right] + \gamma\left(\beta^{-1}\left(1 - \gamma\right) + \gamma/2\right)\|h\|_{\tilde{\mathbf{L}}}^2 \tag{60}$$

we obtain the formulated result

$$\frac{c}{K}\sum_{k=0}^{K-1}\mathbb{E}\left[\left\|\nabla f(x^k)\right\|_{\tilde{\mathbf{L}}^{-1}}^2\right] \le \frac{f(x^0) - \mathbb{E}\left[f(x^K)\right]}{\gamma K} + \left(\beta^{-1}\left(1 - \gamma\right) + \gamma/2\right)\|h\|_{\tilde{\mathbf{L}}}^2. \tag{61}$$

$\square$

**Remark 2.** *1) The first term in the convergence upper bound* (58) *is minimized by maximizing product* $c \cdot \gamma$, *which motivates to choose* $c > 0$ *and* $\gamma \le 1$ *as large as possible. Although due to the constraint on the step size (and* $\beta > 0$)

$$0 < \gamma \le \gamma_{c,\beta} := \frac{1 - c - \beta}{\beta + 1/2}, \tag{62}$$

*constant* $c \in (0, 1)$. *So, by maximizing* $c$ *the value* $\gamma_{c,\beta}$ *becomes smaller, thus there is a trade-off.*

*2) The second term or the neighborhood size (multiplier in front of* $\|h\|_{\tilde{\mathbf{L}}}^2$)

$$\Psi(\beta, \gamma) := \frac{\beta^{-1}(1 - \gamma) + \gamma/2}{c} = \frac{\beta^{-1}(1 - \gamma) + \gamma/2}{1 - \gamma/2 - \beta(1 - \gamma)} \tag{63}$$

*can be numerically minimized (e.g. by using WolframAlpha) with constraints* $\gamma \in (0, 1]$ *and* $\beta > 0$. *The solution of such optimization problem is* $\gamma^\star \approx 1$ *and* $\beta^\star \approx \xi \in \{3.992, 2.606, 2.613\}$. *In fact,* $\Psi(\beta^\star, \gamma^\star) \approx 0.5$.

**Functional gap convergence.** Note that for the quadratic optimization problem (12)

$$\left\| \nabla f(x^k) \right\|_{\tilde{\mathbf{L}}^{-1}}^2 = \left\langle \tilde{\mathbf{L}} x^k - \tilde{\mathbf{b}}, \tilde{\mathbf{L}}^{-1} \left( \tilde{\mathbf{L}} x^k - \tilde{\mathbf{b}} \right) \right\rangle = 2 \left( f(x^k) - f(x^\star) \right). \tag{64}$$

Then by rearranging and subtracting $f^\star := f(x^\star)$ from both sides of inequality (60) we obtain

$$
\begin{aligned}
\mathbb{E}\left[ f(x^{k+1}) \mid x^k \right] - f^\star &\le f(x^k) - f^\star - \gamma c \left\| \nabla f(x^k) \right\|_{\tilde{\mathbf{L}}^{-1}}^2 + \gamma \left( \beta^{-1}(1 - \gamma) + \gamma/2 \right) \|h\|_{\tilde{\mathbf{L}}}^2 \\
&\stackrel{(64)}{=} \left( f(x^k) - f^\star \right) - \gamma c \cdot 2 \left( f(x^k) - f^\star \right) + \gamma \left( \beta^{-1}(1 - \gamma) + \gamma/2 \right) \|h\|_{\tilde{\mathbf{L}}}^2 \\
&= (1 - 2\gamma c) \left( f(x^k) - f^\star \right) + \gamma \left( \beta^{-1}(1 - \gamma) + \gamma/2 \right) \|h\|_{\tilde{\mathbf{L}}}^2.
\end{aligned}
$$

After unrolling the recursion

$$
\begin{aligned}
\mathbb{E}\left[ f(x^{k+1}) \mid x^k \right] - f^\star &\le (1 - 2\gamma c)^k \left( f(x^0) - f^\star \right) + \gamma \left( \beta^{-1}(1 - \gamma) + \gamma/2 \right) \|h\|_{\tilde{\mathbf{L}}}^2 \sum_{i=0}^{k} (1 - 2\gamma c)^i \\
&\le (1 - 2\gamma c)^k \left( f(x^0) - f^\star \right) + \frac{1}{2c} \left( \beta^{-1}(1 - \gamma) + \gamma/2 \right) \|h\|_{\tilde{\mathbf{L}}}^2.
\end{aligned}
$$

This result is formalized in the following Theorem.

**Theorem 5.** *Consider the method* (2) *with estimator* (31) *for a homogeneous quadratic problem* (12) *with positive-definite matrix* $\mathbf{L}_i \equiv \mathbf{L} \succ 0$. *Then if exists* $\mathbf{D}^{-\frac{1}{2}}$ *for* $\mathbf{D} := \text{Diag}(\mathbf{L})$, *scaled permutation sketch* $\mathbf{C}_i' = \sqrt{n} e_{\pi_i} e_{\pi_i}^\top$ *is used and the step size is chosen as*

$$0 < \gamma \le \gamma_{c,\beta} := \frac{1 - c - \beta}{\beta + 1/2}, \tag{65}$$

*where* $\gamma_{c,\beta} > 0$ *for* $\beta + c < 1$. *Then the iterates satisfy*

$$\mathbb{E}\left[ f(x^k) \right] - f^\star \le (1 - 2\gamma c)^k \left( f(x^0) - f^\star \right) + \frac{1}{2c} \left( \beta^{-1}(1 - \gamma) + \gamma/2 \right) \|h\|_{\tilde{\mathbf{L}}}^2, \tag{66}$$

*where* $h = \tilde{\mathbf{L}}^{-1} \tilde{\mathbf{b}} - \frac{1}{\sqrt{n}} \tilde{\mathbf{b}}$ *and* $\tilde{\mathbf{L}} = \mathbf{D}^{-\frac{1}{2}} \mathbf{L} \mathbf{D}^{-\frac{1}{2}}, \tilde{\mathbf{b}} = \mathbf{D}^{-\frac{1}{2}} \mathbf{b}$.

This result shows that for a proper choice of the step size $\gamma = 1$ and constant $c = 1/2$, the functional gap can converge in basically one iteration to the neighborhood of size

$$\|h\|_{\tilde{\mathbf{L}}}^2 = \left\langle \tilde{\mathbf{L}} \left( \tilde{\mathbf{L}}^{-1} \tilde{\mathbf{b}} - \frac{1}{\sqrt{n}} \tilde{\mathbf{b}} \right), \tilde{\mathbf{L}}^{-1} \tilde{\mathbf{b}} - \frac{1}{\sqrt{n}} \tilde{\mathbf{b}} \right\rangle,$$

which equals zero if $\tilde{\mathbf{L}}^{-1} \tilde{\mathbf{b}} = \frac{1}{\sqrt{n}} \tilde{\mathbf{b}}$. This condition is the same as the condition we obtained at the end of Subsection 4.2 with asymptotic analysis of the iterates in the homogeneous case.

**Discussion of the trace.** Consider a positive-definite $\mathbf{L} \succ 0$ such that $\exists \mathbf{D}^{-\frac{1}{2}}$. Thus $\tilde{\mathbf{L}} = \mathbf{D}^{-\frac{1}{2}} \mathbf{L} \mathbf{D}^{-\frac{1}{2}}$ has only ones on the diagonal and $\mathrm{tr}(\tilde{\mathbf{L}}) = n$. Then

$$n \cdot \mathrm{tr}(\tilde{\mathbf{L}}^{-1}) = \mathrm{tr}(\tilde{\mathbf{L}})\mathrm{tr}(\tilde{\mathbf{L}}^{-1}) = (\lambda_1 + \cdots + \lambda_n)\left(\frac{1}{\lambda_1} + \cdots + \frac{1}{\lambda_n}\right) \geq n^2,$$

where the last inequality is due to the relation between harmonic and arithmetic means. Therefore $\mathrm{tr}(\tilde{\mathbf{L}}^{-1}) = \lambda_1^{-1} + \cdots + \lambda_n^{-1} \geq n$ and sum of $\tilde{\mathbf{L}}^{-1}$ eigenvalues has to be greater than $n$.

## C.4 Generalization to $n \neq d$ case.

Our results can be generalized in a similar way as in (Szlendak et al., 2022).

**1)** $d = qn$, for integer $q \geq 1$. Let $\pi = (\pi_1, \ldots, \pi_d)$ be a random permutation of $\{1, \ldots, d\}$. Then for each $i \in \{1, \ldots, n\}$ define

$$\mathbf{C}'_i := \sqrt{n} \cdot \sum_{j=q(i-1)+1}^{qi} e_{\pi_j} e_{\pi_j}^\top. \tag{67}$$

Matrix $\mathbb{E}\left[\overline{\mathbf{B}}^k\right]$ for the homogeneous preconditioned case can be computed as follows:

$$
\begin{aligned}
\mathbb{E}\left[\overline{\mathbf{B}}^k\right] &= \mathbb{E}\left[\frac{1}{n}\sum_{i=1}^{n} \mathbf{C}'_i \tilde{\mathbf{L}} \mathbf{C}'_i\right] \\
&= \frac{1}{n}\sum_{i=1}^{n} \mathbb{E}\left[\sum_{j=q(i-1)+1}^{qi} n e_{\pi_j} e_{\pi_j}^\top \tilde{\mathbf{L}} e_{\pi_j} e_{\pi_j}^\top\right] \\
&= \sum_{i=1}^{n}\sum_{j=q(i-1)+1}^{qi} \mathbb{E}\left[e_{\pi_j} e_{\pi_j}^\top \tilde{\mathbf{L}} e_{\pi_j} e_{\pi_j}^\top\right] \\
&= \sum_{i=1}^{n}\sum_{j=q(i-1)+1}^{qi} \frac{1}{d}\sum_{l=1}^{d} e_l e_l^\top \tilde{\mathbf{L}} e_l e_l^\top \\
&= \sum_{i=1}^{n}\sum_{j=q(i-1)+1}^{qi} \frac{1}{d}\mathrm{Diag}(\tilde{\mathbf{L}}) \\
&= n\frac{q}{d}\mathrm{Diag}(\tilde{\mathbf{L}}) \\
&= \mathrm{Diag}(\tilde{\mathbf{L}}) \\
&= \mathbf{I}.
\end{aligned}
$$

As for the linear term

$$
\begin{aligned}
\mathbb{E}\left[\mathbf{C}' \mathrm{b}\right] &= \mathbb{E}\left[\frac{1}{n}\sum_{i=1}^{n}\mathbf{C}'_i \tilde{\mathrm{b}}\right] = \frac{1}{n}\sum_{i=1}^{n}\mathbb{E}\left[\sum_{j=q(i-1)+1}^{qi}\sqrt{n} e_{\pi_j} e_{\pi_j}^\top \tilde{\mathrm{b}}\right] \\
&= \frac{1}{\sqrt{n}}\sum_{i=1}^{n}\sum_{j=q(i-1)+1}^{qi}\frac{1}{d}\mathbf{I}\tilde{\mathrm{b}} = \frac{\sqrt{n}q}{d}\mathbf{I}\tilde{\mathrm{b}} = \frac{1}{\sqrt{n}}\tilde{\mathrm{b}}.
\end{aligned}
$$

**2)** $n = qd$, for integer $q \geq 1$. Define the multiset $S := \{1, \ldots, 1, 2, \ldots, 2, \ldots, d, \ldots, d\}$, where each number occurs precisely $q$ times. Let $\pi = (\pi_1, \ldots, \pi_n)$ be a random permutation of $S$. Then for each $i \in \{1, \ldots, n\}$ define

$$\mathbf{C}'_i := \sqrt{d} \cdot e_{\pi_i} e_{\pi_i}^\top. \tag{68}$$

$$
\begin{aligned}
\mathbb{E}\left[\overline{\mathbf{B}}^{k}\right] & = \mathbb{E}\left[\frac{1}{n}\sum_{i=1}^{n}\mathbf{C}_i' \tilde{\mathbf{L}}\, \mathbf{C}_i'\right] = \frac{1}{n}\sum_{i=1}^{n}\mathbb{E}\left[d e_{\pi_i} e_{\pi_i}^{\top}\, \tilde{\mathbf{L}}\, e_{\pi_i} e_{\pi_i}^{\top}\right] \\
& = \frac{1}{n}\sum_{i=1}^{n}\frac{1}{d}\sum_{j=1}^{d} d e_j e_j^{\top}\, \tilde{\mathbf{L}}\, e_j e_j^{\top} = \frac{1}{n}\sum_{i=1}^{n}\mathrm{Diag}(\tilde{\mathbf{L}}) = \mathbf{I}.
\end{aligned}
$$

The linear term

$$
\mathbb{E}\left[\mathbf{C}'\,\mathbf{b}\right] = \mathbb{E}\left[\frac{1}{n}\sum_{i=1}^{n}\mathbf{C}_i'\, \tilde{\mathbf{b}}\right] = \frac{1}{n}\sum_{i=1}^{n}\mathbb{E}\left[\sqrt{d} e_{\pi_i} e_{\pi_i}^{\top}\, \tilde{\mathbf{b}}\right] = \frac{\sqrt{d}}{n}\sum_{i=1}^{n}\frac{1}{d}\mathbf{I}\tilde{\mathbf{b}} = \frac{1}{\sqrt{d}}\, \tilde{\mathbf{b}}\,.
$$

To sum up both cases, in a homogeneous preconditioned setting $\mathbb{E}\left[\overline{\mathbf{B}}^{k}\right] = \mathbf{I}$ and

$$
\mathbb{E}\left[\mathbf{C}'\,\mathbf{b}\right] = \mathbb{E}\left[\frac{1}{n}\sum_{i=1}^{n}\mathbf{C}_i'\,\mathbf{b}\right] = \tilde{\mathbf{b}}\,/\sqrt{\min(n,d)}.
$$

Similar modifications and calculations can be performed for heterogeneous scenarios. The case when $n$ does not divide $d$ and vice versa is generalized using constructions from (Szlendak et al., 2022).

## D  COMPARISON TO RELATED WORKS

**Overview of theory provided in the original IST work (Yuan et al., 2022).**  The authors consider the following method

$$x^{k+1} = \mathcal{C}(x^k) - \gamma \nabla f_{i_k}(\mathcal{C}(x^k)), \tag{69}$$

where $[\mathcal{C}(x)]_i = x_i \cdot \mathcal{B}e(p)^4$ is a Bernoulli sparsifier and $i_k$ is sampled uniformly at random from $[n]$.

The analysis in (Yuan et al., 2022) relies on the assumptions

1. $L_i$-smoothness of individual losses $f_i$;
2. $Q$-Lipschitz continuity of $f$: $|f(x) - f(y)| \leq Q\|x - y\|$;
3. Error bound (or PŁ-condition): $\|\nabla f(x)\| \geq \mu\|x^\star - x\|$, where $x^\star$ is the global optimum;
4. Stochastic gradient variance: $\mathbb{E}\left[\|\nabla f_{i_k}(x)\|^2\right] \leq M + M_f \|\nabla f(x)\|^2$;
5. $\mathbb{E}\left[\nabla f_{i_k}(\mathcal{C}(x^k)) \,|\, x^k\right] = \nabla f(x^k) + \varepsilon, \quad \|\varepsilon\| \leq B$.

Convergence result (Yuan et al., 2022, Theorem 1) for step size $\gamma = 1/(2L_{\max})$:

$$\min_{k \in \{1,\dots,K\}} \mathbb{E}\left[\|\nabla f(x^k)\|^2\right] \leq \frac{f(x^0) - f(x^\star)}{\alpha(K+1)} + \frac{1}{\alpha} \cdot \left(\frac{BQ}{2L_{\max}} + \frac{5L_{\max}\omega}{2}\|x^\star\|^2 + \frac{M}{4L_{\max}}\right), \tag{70}$$

where $\alpha := \frac{1}{2L_{\max}}\left(1 - \frac{M_f}{2}\right) - \frac{5\omega L_{\max}}{2\mu^2}$, $\omega := \frac{1}{p} - 1 < \frac{\mu^2}{10L_{\max}^2}$, and $L_{\max} := \max_i L_i$.

If Lipschitzness and Assumption 5 are replaced with *norm condition*:

$$\|\mathbb{E}\left[\nabla f_{i_k}(\mathcal{C}(x^k)) \,|\, x^k\right] - \nabla f(x^k)\| \leq \theta\|\nabla f(x^k)\| \tag{71}$$

they obtain the following (for step size $\gamma = 1/2L_{\max}$)

$$\min_{k \in \{1,\dots,K\}} \mathbb{E}\left[\|\nabla f(x^k)\|^2\right] \leq \frac{f(x^0) - f(x^\star)}{\alpha(K+1)} + \frac{1}{\alpha} \cdot \left(\frac{5L_{\max}\omega}{2}\|x^\star\|^2 + \frac{M}{4L_{\max}}\right), \tag{72}$$

where $\alpha = \frac{1}{2L_{\max}}\left(\frac{1}{2} - \theta - \frac{M_f}{2}\right) - \frac{5\omega L_{\max}}{2\mu^2}$ and $\omega = \frac{1}{p} - 1 < \frac{\mu^2}{5L_{\max}^2\left(\frac{1}{2} - \theta - \frac{M_f}{2}\right)}$.

**Remark 3.** *The original method* (69) *does not incorporate gradient sparsification, which can create a significant disparity between theory and practice. This is because the gradient computed at the compressed model, denoted as $\nabla f(\mathcal{C}(x))$, is not guaranteed to be sparse and representative of the submodel computations. Such modification of the method also significantly simplifies theoretical analysis, as using a single sketch (instead of* **CLC***) allows for an unbiased gradient estimator.*

*Through our analysis of the IST gradient estimator in Equation* (31)*, we discover that conditions— such as Assumption 5 and Inequality* (71)*—are not satisfied, even in the homogeneous setting for a simple quadratic problem. Furthermore, it is evident that such conditions are also not met for logistic loss. At the same time, in general, it is expected that insightful theory for general (non-)convex functions should yield appropriate results for quadratic problems. Additionally, it remains unclear whether the norm condition* (71) *is satisfied in practical scenarios. The situation is not straightforward—even for quadratic problems—as we show in the expression for $\sigma^2$ in Equation* (51)*.*

**Masked training (Mohtashami et al., 2022).**  The authors consider the following "Partial SGD" method

$$\begin{aligned}
\hat{x}^k &= x^k + \delta x^k = x^k - (1 - p) \odot x^k \\
x^{k+1} &= x^k - \gamma p \odot \nabla f(\hat{x}^k, \xi^k),
\end{aligned} \tag{73}$$

---

[4] $\mathcal{B}_p(x) := \begin{cases} x/p & \text{with probability } p \\ 0 & \text{with probability } 1 - p \end{cases}$

where $\nabla f(x, \xi)$ is an unbiased stochastic gradient estimator of a $L$-smooth loss function $f$, $\odot$ is an element-wise product, and $p$ is a binary sparsification mask.

Mohtashami et al. (2022) make the following "bounded perturbation" assumption

$$\max_k \frac{\|\delta x^k\|}{\max\left\{\|p^k \odot \nabla f(x^k)\|, \|p^k \odot \nabla f(\hat{x}^k)\|\right\}} \leq \frac{1}{2L}. \tag{74}$$

This inequality may not hold for a simple convex case. Consider a function $f(x) = \frac{1}{2} x^\top A x$, for

$$A = \begin{pmatrix} a & 0 \\ 0 & c \end{pmatrix}, \qquad x^0 = \begin{pmatrix} x_1 \\ x_2 \end{pmatrix}, \qquad p^0 = \begin{pmatrix} 0 \\ 1 \end{pmatrix}. \tag{75}$$

Then condition (74) (at iteration $k = 0$) will be equivalent to

$$\frac{x_1}{cx_2} \leq \frac{1}{2a} \Leftrightarrow 2 \leq \frac{2a}{c} \leq \frac{x_2}{x_1},$$

which clearly does not hold for an arbitrary initialization $x^0$.

In addition, the convergence bound in (Mohtashami et al., 2022, Theorem 1) suggests choosing the step size as $\gamma_0 \alpha^k$, where

$$\alpha^k = \min\left\{1, \frac{\langle p^k \odot \nabla f(x^k), p^k \odot \nabla f(\hat{x}^k) \rangle}{\|p^k \odot \nabla f(\hat{x}^k)\|^2}\right\} \tag{76}$$

is not guaranteed to be positive to the inner product $\langle p^k \odot \nabla f(x^k), p^k \odot \nabla f(\hat{x}^k) \rangle$, which may lead to non-convergence of the method.

**Optimization with access to auxiliary information framework (Chayti & Karimireddy, 2022)** suggests modeling training with compressed models via performing gradient steps with respect to function $h(x) := \mathbb{E}_{\mathcal{M}}[f(1_{\mathcal{M}} \odot x)]$. This function allows access to a sparse version of the original model $f(x)$. They impose the following bounded Hessian dissimilarity assumption on $h$ and $f$

$$\left\|\nabla^2 f(x) - \mathbb{E}_{\mathcal{M}}\left[\mathbf{D}_{\mathcal{M}} \nabla^2 f(1_{\mathcal{M}} \odot x) \mathbf{D}_{\mathcal{M}}\right]\right\|_2 \leq \delta, \tag{77}$$

where $1_{\mathcal{M}}$ and $\mathbf{D}_{\mathcal{M}} = \text{Diag}(1_{\mathcal{M}})$ refer to a binary vector and matrix sparsification masks.

This approach relies on variance-reduction and requires gradient computations on the full model $x$, and thus it is not suitable for our problem setting.

**Comparison to the work of Liao & Kyrillidis (2022).** Next, we try our best to briefly and accurately represent some of the previous work's findings and comment on the differences.

The authors provide a *high probability* convergence analysis of a "Single Hidden-Layer Neural Network with ReLU activations" based on Neural Tangent Kernel (NTK) framework. The network's first layer weights are initialized based on $\mathcal{N}(0, \kappa^2 \mathbf{I})$ and weight vector of the second layer is initialized uniformly at random from $\{-1, 1\}$. In contrast, we do not make any assumptions on the initialized parameters $x$ (in our notation).

The second differentiation is assumptions on the data. Liao & Kyrillidis (2022) assume that for every data point $(a_j, y_j)$, it holds that $\|a_j\|^2 = 1$ and $|y_j| \leq C - 1$ for some constant $C \geq 1$. Moreover, for any $j \neq l$, it holds that the points $a_i, a_l$ are not co-aligned, i.e., $a_i \neq \xi a_l$ for any $\xi \in \mathbb{R}$. In contrast, we do not make any assumptions about the data apart from the ones on matrices $\mathbf{L}_i$. In addition, analysis by Liao & Kyrillidis (2022) assumes that the number of hidden nodes is greater than a certain quantity and that NN's weights distance from initialization is uniformly bounded.

Liao & Kyrillidis (2022) consider a regression (MSE) loss function, a special case of quadratic loss and full gradients computation. They provide guarantees for IST under a "simplified assumption that every worker has full data access", which corresponds to the homogeneous setting in our terminology.

# E  EXPERIMENTS

To empirically validate our theoretical framework and its implications, we focus on carefully controlled settings that satisfy the assumptions of our work. Specifically, we consider a quadratic problem defined in (12). As a reminder, the local loss function is defined as

$$f_i(x) = \frac{1}{2} x^\top \mathbf{L}_i x - x^\top \mathrm{b}_i,$$

where $\mathbf{L}_i = \mathbf{B}_i^\top \mathbf{B}_i$. Entries of the matrices $\mathbf{B}_i \in \mathbb{R}^{d \times d}$, vectors $\mathrm{b}_i \in \mathbb{R}^d$, and initialization $x^0 \in \mathbb{R}^d$ are generated from a standard Gaussian distribution $\mathcal{N}(0, 1)$.

**Heterogeneous setting.** In Figure 1a, we present the performance of the simplified Independent Subnetwork Training (IST) algorithm (update (2) with estimator (26)) for a heterogeneous problem. We fix the dimension $d$ to 1000 and the number of computing nodes $n$ to 10. We evaluate the logarithm of a relative functional error $\log(f(x^k) - f(x^\star))/(f(x^0) - f(x^\star))$, while the horizontal axis denotes the number of communication rounds required to achieve a certain error tolerance. According to our theory (66), the method converges to a neighborhood of the solution, which depends on the chosen step size. Specifically, a larger step size allows for faster convergence but results in a larger neighborhood.

**Homogeneous setting.** In Figure 1b, we demonstrate the convergence of the iterates $x^k$ for a homogeneous problem with $d = n = 50$. The results are in close agreement with our theoretical predictions for the estimator (31). We observe that the distance to the method's expected fixed point $x^\infty = \tilde{\mathrm{b}} / \sqrt{n}$ decreases linearly for different step size values. This confirms that IST may not converge to the optimal solution $x^\star = \tilde{\mathbf{L}}^{-1} \tilde{\mathrm{b}}$ of the original problem (12) in general (no interpolation) cases. In addition, there are no visible oscillations in comparison to the heterogeneous case.

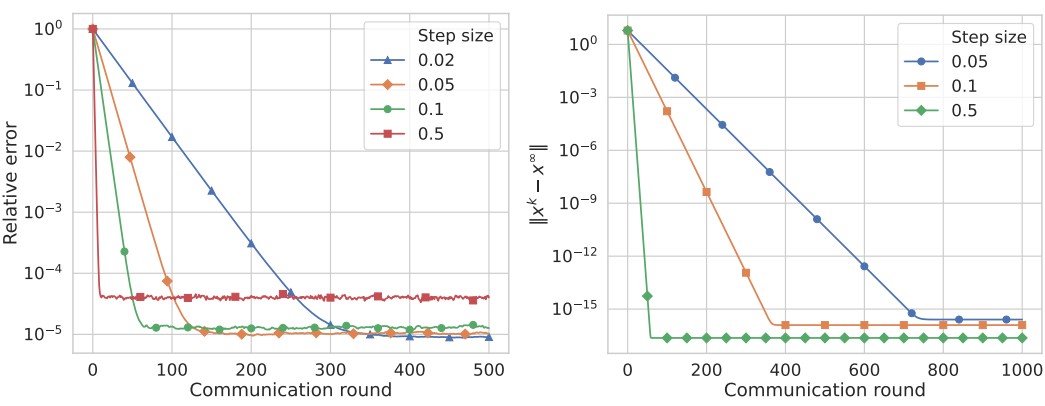

(a) Function convergence for heterogeneous case.    (b) Iterates convergence for homogeneous case.

Figure 1: Performance of simplified IST on a quadratic problem for varying step size values.

Simulations were performed on a machine with $24 \, \mathrm{Intel(R)} \, \mathrm{Xeon(R)} \, \mathrm{Gold} \, 6246 \, \mathrm{CPU} \, @ \, 3.30 \, \mathrm{GHz}$.

