# OpenReview forum: "Towards a Better Theoretical Understanding of Independent Subnetwork Training"
_ICLR.cc/2024/Conference — Submitted to ICLR 2024_

### Official Review · Reviewer_2HE3 · 2023-10-25

**Soundness:** 3 good
**Presentation:** 2 fair
**Contribution:** 2 fair
**Rating:** 5
**Confidence:** 3

**Summary:**

> **TL;DR:** The paper provides a theoretical analysis on the IST algorithm. The analysis is more flexible than previous IST theoretical analysis works. However, I find the experimental work lacking. Addressing my concerns and questions can improve the score.

The paper presents a comprehensive analysis of Independent Subnetwork Training (IST) in the context of distributed machine learning with a focus on data and model parallelism. The study identifies the lack of a rigorous understanding of IST convergence as a motivation for the research. The main contributions of this work include a novel approach to analyzing distributed methods that combine data and model parallelism, an analysis of IST in both homogeneous and heterogeneous scenarios without restrictive gradient estimator assumptions, and the identification of settings where IST can optimize efficiently or converge to a well-characterized irreducible neighborhood. The research is supported by carefully designed experiments and provides valuable insights into the advantages and limitations of IST in large-scale machine learning.

**Strengths:**

* **S.1.** The paper provides an in-depth analysis on the IST algorithm which tackles an important problem.
* **S.2.** The paper provides a theoretical analysis with higher flexibility.
* **S.3.** The theoretical analysis includes both the Homogeneous and the Heterogeneous settings.

**Weaknesses:**

* **W.1.** The provided experimental results are not conclusive enough and placed in the end of the Appendix.
* **W.2.** The work is mainly focused on the quadratic model.

**Questions:**

* **Q.1.** Can this work be extended to neural networks such as [1]?

[1] Dun, Chen, Cameron R. Wolfe, Christopher M. Jermaine, and Anastasios Kyrillidis. "Resist: Layer-wise decomposition of resnets for distributed training." In Uncertainty in Artificial Intelligence, pp. 610-620. PMLR, 2022.

---

> ### Author Response · Authors · 2023-11-13
> **Response to Reviewer 2HE3**
>
> Dear Reviewer 2HE3,
>
> Thanks for the time and effort devoted to our paper.
>
> ## Comments on Weaknesses
>
> > The provided experimental results are not conclusive enough and placed in the end of the Appendix.
>
> We comment on this in our general comment ([link](https://openreview.net/forum?id=HhVns87e74&noteId=Y3PkEuz0tL)) on experiments to all reviewers.
>
> We would like to highlight that the title of our work is _“Towards a Better **Theoretical** Understanding of IST”_. That is why, in the main part of the paper, we focus on contributions from theoretical analysis. Another reason is the limited space allowed by the conference submission format. If the reviewer believes that moving the experimental results to the main part will strengthen the points we are making, we can do this in the camera-ready version.
>
> > The work is mainly focused on the quadratic model.
>
> We address this in our general response ([link](https://openreview.net/forum?id=HhVns87e74&noteId=fjCbL9Q2IZ)) to all reviewers regarding assumptions.
>
> This model is chosen for its commonality in supervised machine learning, including its use in neural networks and its demonstrated efficacy in providing theoretical insights for complex optimization algorithms. In our paper, we emphasize that, despite its apparent simplicity, the quadratic model poses significant analytical challenges, particularly due to the biased gradient estimator used in IST. This makes it a compelling choice for analyzing complex phenomena within optimization algorithms. We further demonstrated that even in seemingly straightforward scenarios, such as the homogeneous interpolation case, the algorithm might not converge.
>
> ## Responses to questions
>
> > Can this work be extended to neural networks such as [1]?
>
> As far as we know, convergence of ResNets training is not adequately understood even for much simpler optimization algorithms such as gradient descent. This lack of understanding is compounded in the case of IST due to the biased nature of the gradient estimator and the complexities inherent in distributed training. Given these challenges, a  straightforward extension of our current work to ResNets or similar architectures is not feasible at this stage.
>
>
> Best regards, Authors

---

### Official Review · Reviewer_cu8b · 2023-10-31

**Soundness:** 3 good
**Presentation:** 3 good
**Contribution:** 2 fair
**Rating:** 6
**Confidence:** 3

**Summary:**

This paper provides a theoretical convergence analysis of the independent subnetwork training (IST) method on a quadratic model.

**Strengths:**

1. As a theoretical paper, the paper is well-organized and easy to follow. Proofs and more details are attached in the appendices.
2. This paper presents a theoretical analysis for the recent IST method.

**Weaknesses:**

### Major issues
1. Section 2.2 lists 3 assumptions for the theoretical analysis. The authors also discuss the necessity of each assumption. Could the authors try to remove one of them? Specifically, is it possible to discuss other problems other than the specific quadratic one?
2. The paper focuses on the theoretical aspects of IST. It would be insightful to discuss the practical implications of the findings for real-world applications and provide guidance on effectively utilizing IST in various distributed training scenarios.

### Minor issues
1. At the bottom of Page 2, $\mathbf{R}^d \rightarrow \mathbf{R}$
2. Figures 1a and 1b have different vertical axis, relative error and absolute error. Could the authors provide both relative and absolute error for these two cases.

**Questions:**

What are the limitations and potential negative impacts of the paper?

---

> ### Author Response · Authors · 2023-11-13
> **Response to Reviewer cu8b**
>
> Dear Reviewer cu8b,
>
> Thanks for the time and effort devoted to our paper. We greatly value careful reading of the material and the positive evaluation of our work.
>
> ## Comments on Weaknesses
>
> > Section 2.2 lists 3 assumptions for the theoretical analysis. The authors also discuss the necessity of each assumption. Could the authors try to remove one of them? Specifically, is it possible to discuss other problems other than the specific quadratic one?
>
> We address this weakness in our general response ([link](https://openreview.net/forum?id=HhVns87e74&noteId=fjCbL9Q2IZ)) to all reviewers regarding assumptions.
>
> We note that the assumption of exact submodel gradient computation can be easily generalized to stochastic estimators with bounded variance. Regarding the choice of a quadratic model. We chose this approach due to the clear theoretical insights it provides, as detailed in our study. Analyses of similar methods with non-quadratic loss functions have often led to restrictive and impractical assumptions, resulting in unsatisfactory convergence bounds, as discussed in Section 4.3 and Appendix C. The complexity and lack of a suitable theoretical framework for a general class of $L$-smooth functions, particularly with challenges like biased gradient estimators, further motivated our decision.
>
> While we acknowledge the value of extending our analysis to non-quadratic settings, the current theoretical landscape and the intricacies of such an extension necessitated our initial focus on the quadratic model. This foundational work sets the stage for future exploration in more complex loss functions as the field advances.
>
>
> > The paper focuses on the theoretical aspects of IST. It would be insightful to discuss the practical implications of the findings for real-world applications and provide guidance on effectively utilizing IST in various distributed training scenarios.
>
> Thank you for your note on the practical implications of our findings. We discussed it in our general comment ([link](https://openreview.net/forum?id=HhVns87e74&noteId=Y3PkEuz0tL)) on experiments to all reviewers.
>
> Briefly, our study indicates that IST can be highly efficient in both homogeneous and heterogeneous settings, given interpolation conditions typical for large neural networks. However, in the more general case, we found that the convergence of naïve IST is influenced by the level of heterogeneity among the distributed nodes. Specifically, decreasing the learning rate can be especially helpful in heterogeneous scenarios for reducing error.
>
>
> > Figures 1a and 1b have different vertical axis, relative error and absolute error. Could the authors provide both relative and absolute error for these two cases.
>
> We would like to note that it was done on purpose. The first (1a) plot illustrates that the method (functional gap) converges to the neighborhood of the solution in the first case. While the second (1b) one shows that the method’s iterates converge to a fixed point different from the optimum. We can provide additional results in the camera-ready version of the paper.
>
> > At the bottom of Page 2, $\mathbf{R}^d \to \mathbf{R}$
>
> Thank you for pointing this out. We already fixed it in our revision.
>
> Best regards, Authors

---

### Official Review · Reviewer_GaQK · 2023-11-04

**Soundness:** 2 fair
**Presentation:** 3 good
**Contribution:** 3 good
**Rating:** 5
**Confidence:** 3

**Summary:**

This paper tries to provide a theoretical understanding of IST’s optimization performance using a quadratic model under no restrictive/specific assumptions on sparsifiers. Both homogeneous and heterogeneous scenarios are discussed, and the latter one is closer to practical scenarios. It provides insights into when IST can optimize very efficiently or not converge to the optimal
solution with tight characterization.

**Strengths:**

* This paper develops a more rigorous theoretical analysis of IST convergence, although with a simple quadratic model.
* The paper is overall well-written and easy to follow. The assumptions are stated explicitly, and the notations are mostly clear.
* Identify the settings IST may not converge, which hopefully can have implications for NN training.

**Weaknesses:**

* **Restrictive (maybe impractical) assumption the work uses that assumes performing only one gradient descent step during local training in IST**. Performing only one gradient computation at each node requires very frequent communication with the server, which will incur high communication costs. Hence, performing multiple gradient descent steps in the local mode is more desired as in the IST work [1] and [2]. I admit taking multiple steps may hurt the accuracy, while the main motivation of IST is saving communication costs under the accuracy-efficiency trade-off.
*  **Why is the gradient sparsification introduced in Eq.6 for IST?** The original IST work formulated the IST training method as Eq(69)  without the gradient sparsification operator. The authors state that “it can create a significant disparity between theory and practice” in Appendix D. I didn’t notice that IST uses any form of gradient sparsification, as IST is orthogonal to gradient sparsification techniques.  The motivation for introducing this operator in IST formulation is unclear.

[1] Binhang Yuan, Cameron R Wolfe, Chen Dun, Yuxin Tang, Anastasios Kyrillidis, and Chris Jermaine. Distributed learning of fully connected neural networks using independent subnet training. Proceedings of the VLDB Endowment, 15(8):1581–1590, 2022.

[2] Chen Dun, Cameron R Wolfe, Christopher M Jermaine, and Anastasios Kyrillidis. ResIST: Layerwise decomposition of resnets for distributed training. In Uncertainty in Artificial Intelligence, pp. 610–620. PMLR, 2022.

**Questions:**

* Can the author further discuss why the assumption that performing only one gradient computation is reasonable, as it is not a common choice for efficient IST? It would be better if the author could show the theoretic analysis still holds with a more relaxed assumption, like performing twice or more.
* Can the authors provide a clear discussion on the reason for introducing the gradient sparsification operator compared to the original formula? Why is this operator necessary when analyzing the convergence of IST if those two techniques are orthogonal?

---

> ### Author Response · Authors · 2023-11-13
> **Response to Reviewer GaQK**
>
> Dear Reviewer GaQK,
>
> Thanks for the time and effort devoted to our paper. We greatly value careful reading of the material and Appendix.
>
> ## Responses to questions
>
> > Can the author further discuss why the assumption that performing only one gradient computation is reasonable, as it is not a common choice for efficient IST? It would be better if the author could show the theoretic analysis still holds with a more relaxed assumption, like performing twice or more.
>
> We address this comment in our general response ([link](https://openreview.net/forum?id=HhVns87e74&noteId=fjCbL9Q2IZ)) to all reviewers regarding assumptions.
>
> We understand that this assumption might seem restrictive and potentially impractical due to the increased communication costs it could entail. However, it is required to isolate and analyze specific properties of IST more effectively. We would like to highlight that our theory could potentially be extended to scenarios involving multiple local steps, as suggested in the approach of Khaled et al. (2019). However, the current state of research indicates that no conclusive analysis demonstrates the advantages of multiple local steps in the worst-case scenarios. We believe that the insights gained from our approach are crucial for guiding future research that might explore more complex scenarios, including those involving multiple local steps.
>
>
> > Can the authors provide a clear discussion on the reason for introducing the gradient sparsification operator compared to the original formula? Why is this operator necessary when analyzing the convergence of IST if those two techniques are orthogonal?
>
> Thank you for bringing this up! Let us clarify the point.
>
> Gradient sketching is introduced to better represent submodel computations. Note that when the gradient is taken with respect to the submodel $\nabla f(\mathbf{C} x)$ it can result in a non-sparse update, which contradicts the idea of IST. To illustrate this, imagine a logistic regression problem, for which computing a gradient for compressed weights vector (even for zero) may result in a non-sparse gradient (potentially without any zeros at all). That is why we believe that sparsification of the gradient is an essential component of correclty representing IST. Thus, our work improves upon the analysis of the original paper.
>
>
> Best regards, Authors
>
> ___
>
> Khaled, Ahmed, et al. "First analysis of local GD on heterogeneous data." arXiv preprint:1909.04715 (2019).

---

### Official Review · Reviewer_VkZR · 2023-11-05

**Soundness:** 3 good
**Presentation:** 3 good
**Contribution:** 2 fair
**Rating:** 5
**Confidence:** 2

**Summary:**

This paper proposes a theoretical analysis framework to understand the behavior of independent subnetwork training. Expanding previous work, this work enables analysis of model parallelism, which is widely used for training massive-scale neural network models. The authors analyze homogeneous and heterogeneous scenarios and suggest settings for efficient convergence. The authors provide limited experimental support to validate their analysis.

**Strengths:**

- First in class to analyze distributed training scenarios for a better understanding of their success and failure, beyond data parallelism.

- Explain in detailed procedures for establishing the analysis framework for independent subnetwork training

**Weaknesses:**

- Theoretical understanding seems to be constrained by the assumptions, which might separate the current analysis from the real use cases

- Although the suggested analysis of the convergence and bias sounds interesting and useful, the limited experimental validation would limit the application of the proposed observation in real distributed training scenarios. In particular, the authors have emphasized the need for the theoretical understanding of a wide-spread parallelization and co-design of communication and training algorithms for large-scale training, but the limited validation would hinder the application of the findings from this work.

**Questions:**

Can we see a more realistic distributed training scenario (e.g., training ImageNet deep neural networks) to validate the key observations of this work?

---

> ### Author Response · Authors · 2023-11-13
> **Response to Reviewer VkZR**
>
> Dear Reviewer VkZR,
>
> Thanks for your time and effort.
>
>
> ## Comments on Weaknesses
>
> > Theoretical understanding seems to be constrained by the assumptions, which might separate the current analysis from the real use cases
>
> We address this comment in our general response ([link](https://openreview.net/forum?id=HhVns87e74&noteId=fjCbL9Q2IZ)) to all reviewers regarding assumptions.
>
> We want to emphasize that in our study, the algorithm and problem setting are intentionally simplified to precisely isolate and understand the unique effects of combining data with model parallelism, especially in the context of distributed submodel gradient computations. We are aware that assumptions we make may deviate from practical settings. However, our goal was not to replicate real-world scenarios closely but to develop a fundamental understanding of certain theoretical aspects of IST.
>
> > Although the suggested analysis of the convergence and bias sounds interesting and useful, the limited experimental validation would limit the application of the proposed observation in real distributed training scenarios.
>
> We comment on this Weakness in our general comment ([link](https://openreview.net/forum?id=HhVns87e74&noteId=Y3PkEuz0tL)) on experiments to all reviewers.
>
> In addition, let us stress that the main contribution of our work is from the theory side, and that is why, in our experiments, we focus on well-controlled settings that satisfy the assumptions in our paper to provide evidence that our theory translates into observable predictions. These are well-designed experiments that do support our theory and core claims. Since our results guarantee that the methods work, we do not need to test them extensively on large or complicated datasets and models to show that they do (which is necessary for heuristics unsupported by any theory).
>
> ## Responses to questions
>
> > Can we see a more realistic distributed training scenario (e.g., training ImageNet deep neural networks) to validate the key observations of this work?
>
> We consider extending our experimental results with neural network training for the camera-ready version of the paper. Due to time constraints, it is not possible to provide them during the rebuttal period.
>
>
> Best regards, Authors

---

### Author Response · Authors · 2023-11-13
**General message**

Dear Program Committee Members,

We thank the reviewers for their time reading our paper and for their feedback. This is all much appreciated!

We are heartened to note that the problem addressed in our paper is recognized as _"important"_ to tackle (Reviewer 2HE3). The descriptions of our _"in-depth"_ (Reviewer 2HE3) analysis as _"First in class"_ (Reviewer VkZR), _"more rigorous"_ (Reviewer GaQK), and possessing _"higher flexibility"_ (Reviewer 2HE3) are particularly encouraging. Furthermore, we are pleased that Reviewers cu8b31 and GaQK find our paper to be _"well-organized"_, _"well-written"_, and _"easy to follow "_. Such recognition of the paper's strengths motivates us to refine our work further.

We look forward to a constructive discussion and are open to any further questions you may have. Please do not hesitate to point out any aspects you find unconvincing or require additional clarification.

Best regards,

Paper 5130 Authors

---

### Author Response · Authors · 2023-11-13
**General response regarding assumptions**

In our work, we make the following simplifications:

1. Exact submodel gradient computation
2. Single local step / gradient computation
3. Quadratic model

Please note that we made these assumptions with a particular purpose in mind. Our goal was not to try to analyze the closest to a practical setting problem but rather to focus on specific new (and challenging) properties of the considered formulation. Quite often, such "closest to a practical" approaches, unfortunately, lead to very loose (sometimes even vacuous) bounds, which are obtained under restrictive (problematic to check) assumptions introduced to facilitate the analysis, such as, i.e., bounded gradient norm and strong convexity (almost conflicting conditions). In the paper, we discuss this in detail by comparing it to prior works that tried to analyze similar settings.

We believe that the contribution of this work includes a formalization of a novel theoretical setting, which basically has not been studied before. That is why it was essential for us to "isolate" the effects of homogeneous/heterogeneous distribution and computations with respect to submodels. To illustrate our position, let us refer to breakthrough optimization works on clipping [1] and local methods [2] (in addition to those mentioned in our paper works on GD with delayed updates and cyclical step-sizes), which considered simple (full) gradient descent updates which allowed to focus on the particular challenges of the considered problem formulations and provided insights that led to an improved understanding of the corresponding methods.


___


[1] Zhang, Jingzhao, et al. "Why Gradient Clipping Accelerates Training: A Theoretical Justification for Adaptivity." ICLR 2019.

[2] Khaled, Ahmed, et al. "First analysis of local GD on heterogeneous data." arXiv preprint:1909.04715 (2019).

[3] Woodworth, Blake E., et al. "Minibatch vs local SGD for heterogeneous distributed learning." NeurIPS 2020.

---

> ### Author Response · Authors · 2023-11-13
> **Continuation of the general response regarding assumptions**
>
> Next, we comment on the necessity and reasons for each simplification, expanding on the paper's subsection 2.2.
>
> ### 1)  Exact submodel gradient computation
>
> Our results **can be easily generalized** to unbiased stochastic gradient estimators $g(x)$ with a bounded variance (one of the most used in the stochastic optimization literature)
> $$\mathbb{E} \||g(x) - \mathbb{E} g(x) \||^2 \leq \delta^2.$$
> Namely, such local gradient estimators will introduce an additional neighborhood term $\gamma \delta^2$ in our convergence upper bounds. This can be obtained using the bias-variance decomposition:
> $$
> \mathbb{E} \||g(x)\||^2 = \mathbb{E} \||g(x) - \mathbb{E} g(x) \||^2 + \|| \mathbb{E} g(x) \||^2.
> $$
>
> ### 2) Single local step / gradient computation
>
> Our theory can be extended to methods taking multiple local steps using the approach of, e.g., [2]. However, we **deliberately avoided** this as it would lead to worse theoretical results, as simple local methods (like local SGD) were shown to be dominated by simpler non-local counterparts in the heterogeneous setting [3]. So, even after several years of studying local update algorithms (at least since 2019), no satisfactory analysis shows its benefits in the worst case.
>
> ### 3) Quadratic model
> To start, quadratic loss function is one of the most common (along with cross-entropy) choices of loss functions in supervised machine learning. Neural networks are not an exception.
>
> We want to bring your attention to the fact that even for such a “simple” problem, the **analysis is very non-trivial** because the gradient estimator of the studied method is biased. We show that even for an interpolation homogeneous case, the algorithm may not converge. Our main goal is to study the properties of the method used for training large models with combined model and data parallelism. We believe that the quadratic problem is very well demonstrative for our purposes. In addition, as of today, we are not aware of any optimization theory for deep neural networks that does not simplify the actual practical settings.
>
> There were already attempts to perform analysis of similar classes of methods for a different class of loss functions. There is a discussion of their results in Section 4.3 and Appendix D in more detail. We find their convergence bounds unsatisfying from a theoretical viewpoint due to too restrictive additional assumptions (e.g., on bounded gradient norm or sparsification parameter), which lead to vacuous bounds in some instances. That is why we decided to take a step back and start with a quadratic problem setting, which allowed us to perform a precise theoretical analysis and reveal the advantages and limitations of IST. Generalizing our results to a non-quadratic setting is currently an open and potentially challenging problem. When we tried to solve this issue ourselves, it was found that we are not aware of a theoretical framework that allows to perform analysis for a general class of $L$-smooth functions due to challenges (e.g., biased gradient estimator) mentioned in Section 2.1 and before 4.3. We would greatly appreciate it if the reviewers shared any ideas on how to extend our theoretical results.

---

### Author Response · Authors · 2023-11-13
**General comment on experiments**

We appreciate the feedback regarding the experimental aspects of our paper. We would like to clarify our approach and the main objectives of our work, which predominantly lie in the theoretical domain.

Our primary goal is to establish a rigorous mathematical framework for understanding the Independent Subnetwork Training (IST), particularly its convergence properties. This study represents a pioneering effort in comprehensively analyzing the IST algorithms that integrate model and data parallelism, an area that has scarcely been explored.

Our theoretical insights shed light on both the strengths and limitations of IST in various scenarios:

- In the **interpolation** case (often considered applicable to modern Deep Learning models), IST can perform very efficiently in both homogeneous and heterogeneous scenarios.

- In the more **general** case, we show that naïve/vanilla IST’s convergence heavily depends on the heterogeneity level, which may require decreasing the step size later throughout the optimization. While in a data center (local cluster) setting, this can be fixed by access to shared data, in federated learning (very heterogeneous), a different method may be needed.

We believe that to guide the future design of IST, first, it is essential to understand its primary and fundamental properties. The experiments we conducted are intentionally designed to align with our theoretical assumptions, providing tangible evidence that our predictions hold true in controlled settings. These experiments support our core theoretical claims.

Given the theoretical nature of our contribution, extensive testing on large or complex datasets was not within the scope of this work. Our aim was not to establish new practical benchmarks but to lay a foundational understanding of IST’s theoretical properties. Numerous studies, as cited in our introduction, focus on the experimental validation of IST. Our work, in contrast, addresses the critical gap in theoretical comprehension of training methods with compressed models.

---

### Meta-Review · Area_Chair_Na9R · 2023-12-06

**Metareview:**

This paper proposed a theoretical framework for understanding the behavior of independent subnetwork training (IST) in distributed machine learning. It is recognized for its novel approach in analyzing distributed training scenarios and providing a more rigorous theoretical analysis of IST convergence. The paper is well-organized, easy to follow, and the inclusion of both homogeneous and heterogeneous scenarios is appreciated. The analysis extends beyond data parallelism, which is a significant contribution to the field.

However, a major concern raised by reviewers is the constrained theoretical understanding due to the assumptions made, which might not align with real-world use cases. The experimental validation is limited, hindering the application of findings to practical distributed training scenarios. Reviewers raised concerns about the practicality of the assumptions, such as performing only one gradient descent step during local training in IST, which may not be efficient in real scenarios. There are also questions about the introduction of the gradient sparsification operator in the IST formulation, which seems unclear and potentially unnecessary. Besides, the paper's focus on a quadratic model, though understandable as a theory foundation paper, limits its applicability (not a major weakness).

While the theoretical aspects are strong, the gaps in practical applicability and validation make this paper fall short of the acceptance threshold. The authors are encouraged to revise towards the next venue.

**Justification For Why Not Higher Score:**

See above

**Justification For Why Not Lower Score:**

N/A

---

### Decision · Program_Chairs · 2024-01-16

Reject